# Non-monotone Submodular Optimization: $p$-Matchoid Constraints and Fully Dynamic Setting

**Kiarash Banihashem**[*]
University of Maryland
College Park, MD, USA
kiarash@umd.edu

**Samira Goudarzi**[*]
University of Maryland
College Park, MD, USA
samirag@umd.edu

**MohammadTaghi Hajiaghayi**[*]
University of Maryland
College Park, MD, USA
hajiagha@umd.edu

**Peyman Jabbarzade**[*]
University of Maryland
College Park, MD, USA
peymanj@umd.edu

**Morteza Monemizadeh**[*]
TU Eindhoven
Eindhoven, The Netherlands
M.Monemizadeh@tue.nl

## Abstract

Submodular maximization subject to a $p$-matchoid constraint has various applications in machine learning, particularly in tasks such as feature selection, video and text summarization, movie recommendation, graph-based learning, and constraint-based optimization. We study this problem in the dynamic setting, where a sequence of insertions and deletions of elements to a $p$-matchoid $\mathcal{M}(\mathcal{V}, \mathcal{I})$ occurs over time and the goal is to efficiently maintain an approximate solution. We propose a dynamic algorithm for non-monotone submodular maximization under a $p$-matchoid constraint. For a $p$-matchoid $\mathcal{M}(\mathcal{V}, \mathcal{I})$ of rank $k$, defined by a collection of $m$ matroids, our algorithm guarantees a $(2p + 2\sqrt{p(p+1)} + 1 + \epsilon)$-approximate solution at any time $t$ in the update sequence, with an expected amortized query complexity of $O(\epsilon^{-3} p k^4 \log^2(k))$ per update.

## 1 Introduction

A $p$-matchoid $\mathcal{M}(\mathcal{V}, \mathcal{I})$ consists of $m$ matroids $\mathcal{M}_1(\mathcal{V}_1, \mathcal{I}_1), \mathcal{M}_2(\mathcal{V}_2, \mathcal{I}_2), \ldots, \mathcal{M}_m(\mathcal{V}_m, \mathcal{I}_m)$, where each element in the ground set $\mathcal{V}$ appears in the ground sets of at most $p$ of these matroids. Additionally, every subset $X \subseteq \mathcal{V}$ is an independent set in $\mathcal{I}$ if $X \cap \mathcal{V}_i \in \mathcal{I}_i$ for any matroid $\mathcal{M}_i(\mathcal{V}_i, \mathcal{I}_i)$. In the context of submodular maximization subject to a $p$-matchoid constraint $\mathcal{M}(\mathcal{V}, \mathcal{I})$, the objective is to find an independent set $I^* \in \mathcal{I}$ that maximizes $f(I^*)$, where $f : 2^{\mathcal{V}} \to \mathbb{R}^{\geq 0}$ is a (non-monotone) submodular function defined over subsets of the ground set $\mathcal{V}$.

The $p$-matchoid constraint has various applications in machine learning, particularly in tasks involving optimization and constraint satisfaction including the followings:

**Feature selection:** In feature selection, where the goal is to pick a subset of features that works well while keeping things simple, $p$-matchoids can be used to model constraints on the relationships between features. For example, if certain features are highly correlated or redundant, a $p$-matchoid constraint can ensure that only a limited number of such features are chosen together Das and Kempe [2008], Khanna et al. [2017], Elenberg et al. [2016], Qian and Singer [2019], Quinzan et al. [2023].

**Video summarization and movie recommendation tasks:** Submodular maximization under $p$-matchoid constraints is crucial for video summarization Feldman et al. [2018], Mirzasoleiman et al. [2018], Gu et al. [2023] and movie recommendation Harshaw et al. [2022], Badanidiyuru et al. [2020],

---

[*]Equal Contribution

Liu et al. [2022], Tschiatschek et al. [2018] tasks. It helps select representative content and generate personalized and diverse recommendations, leading to improved user experience and engagement.

**Text summarization:** In text summarization Lin and Bilmes [2011], Bairi et al. [2015], Lin and Bilmes [2010], Liu et al. [2013], summarizing multiple documents together (e.g., news articles on the same topic or a survey paper) is often necessary. In such scenarios, $p$-matchoids help ensure that the resulting summary represents a balanced combination of information from each document.

**Viral marketing campaigns:** Viral marketing promotes products or services through social sharing and word-of-mouth for rapid, exponential message spread. $p$-matchoids can model relationships and interactions, capturing constraints on information dissemination and influence propagation Zhang et al. [2023], Jin et al. [2021], Cui et al. [2021], Kempe et al. [2003].

In general, $p$-matchoids provide a flexible way to add complex rules to optimization problems in machine learning. By setting $p$-matchoid constraints, practitioners can adjust optimization algorithms to meet specific requirements and constraints, resulting in more robust models Quinzan et al. [2021], Friedrich et al. [2019], Wei et al. [2015].

In this paper, we study the non-monotone submodular maximization problem under a $p$-matchoid constraint in the *fully dynamic* setting. This model considers a sequence of updates, each of which is either an insertion or a deletion. Using the notation $V_t \subseteq \mathcal{V}$ to represent the set of elements present at any given time $t$, this model assumes that $V_t$ is derived from $V_{t-1}$ after the insertion or deletion of an element $e$ at time $t$.

The ultimate objective of this problem is to maintain an independent set $I^* \in \mathcal{I}$ at each time $t$, such that $f(I^*) = OPT_t$, where $OPT_t = \max_{I \in \mathcal{I}, I \subseteq V_t} f(I)$ is the optimal submodular value among any independent set in $\mathcal{I}$ present at time $t$. However, finding an exact solution for this problem, even in an offline setting where the elements are fixed and provided at once, is NP-hard as it generalizes problems such as unconstrained non-monotone submodular maximization and monotone submodular maximization under a cardinality constraint, both of which are known to be NP-hard Feige et al. [2011], Nemhauser et al. [1978]. Therefore, we focus on developing approximate dynamic algorithms and use their approximation ratios to evaluate their quality. Also, following prior work, we assume having an oracle access to the submodular function $f$, and measure the performance of our algorithms based on their query complexities.

To underscore some specific aspects of this problem that make it more challenging than some other similar problems in the literature, we highlight the following:

- The nature of the $p$-matchoid constraint, which is extremely versatile and generalizes even complex combinatorial structures such as the intersection of $p$ matroids, $p$-uniform hypergraph matching, and by extension, matroid, and cardinality constraints, making it significantly more complex to handle.

- Function $f$ not being monotone, as non-monotone submodular maximization is inherently more difficult than its monotone counterpart, evidenced by the weaker approximation guarantees and increased complexity it typically yields across different settings and constraints, adding to the challenge.

- The dynamic setting of the problem, where elements get inserted or deleted adversarially, and it requires algorithms to adapt to real-time changes.

To date, the most significant advancement in non-monotone submodular maximization in the dynamic setting is a result presented at NeurIPS'23. In their pioneering study, Banihashem, Biabani, Goudarzi, Hajiaghayi, Jabbarzade, and Monemizadeh Banihashem et al. [2023] proposed a dynamic algorithm for non-monotone submodular maximization under a cardinality constraint $k$. Their algorithm achieves a $(8 + \epsilon)$-approximation while requiring an expected amortized $O(\epsilon^{-2} k^2 \log^3(k))$ oracle queries per update.

Despite this progress, the development of dynamic algorithms for non-monotone submodular maximization under a *matroid constraint*, or more generally a *p-matchoid constraint*, remains an open and unexplored challenge, one that we address in this study.

## 1.1 Our contribution and techniques

We present the main result of this paper in the following theorem.

**Theorem 1.1** (Main Theorem). *Let $0 < \epsilon \leq 1$ be a parameter. Let $\mathcal{M}(\mathcal{V}, \mathcal{I})$ be a p-matchoid consisting of m matroids $\mathcal{M}_1(\mathcal{V}_1, \mathcal{I}_1), \mathcal{M}_2(\mathcal{V}_2, \mathcal{I}_2), \ldots, \mathcal{M}_m(\mathcal{V}_m, \mathcal{I}_m)$, and let k denote the size of the largest independent set in $\mathcal{I}$. Let $f : 2^{\mathcal{V}} \to \mathbb{R}^{\geq 0}$ be a (not necessarily monotone) submodular function defined over subsets of the ground set $\mathcal{V}$. Then, there exists a dynamic algorithm for maximizing the function f subject to the p-matchoid constraint $\mathcal{M}(\mathcal{V}, \mathcal{I})$, which maintains a $(2p+2\sqrt{p(p+1)}+1+\epsilon)$-approximate solution at any time t during the sequence of updates, while performing an expected amortized $O(\epsilon^{-3}pk^4 \log^2 k)$ oracle queries per update.*

As a byproduct, we obtain another result for the special case of cardinality k, as stated below.

**Corollary 1.2.** *There exists a dynamic algorithm for non-monotone submodular maximization subject to a matroid constraint of rank k (including the cardinality constraint k), which achieves a $(5.82 + \epsilon)$-approximate solution using $O(\epsilon^{-3}k^4 \log^2 k)$ oracle queries per update.*

Our algorithm for the cardinality case is an improvement upon the approximation of the dynamic algorithm presented by Banihashem et al.Banihashem et al. [2023], which maintains a $(8 + \varepsilon)$-approximate solution with an expected amortized $O(\varepsilon^{-2}k^2 \log^3(k))$ oracle queries per update.

Under the assumption that the submodular function $f$ is monotone, the approximation guarantee of Theorem 1.1 can be strengthened. An adaptation of our algorithm then yields the following side result:

**Proposition 1.3.** *There exists a dynamic algorithm for monotone submodular maximization subject to a p-matchoid constraint of rank k with $(4p + \epsilon)$ approximation guarantee and $O(\varepsilon^{-3}pk^4 \log^2(k))$ update time.*

### 1.1.1 Overview of our algorithm

We present a dynamic algorithm for non-monotone submodular maximization under $p$-matchoid constraints, where the ground set evolves through a sequence of insertions and deletions. Our approach builds on an offline recursive framework that incrementally constructs a feasible solution by combining two core ideas: filtering and selective sampling. The offline algorithm processes elements in multiple levels, iteratively refining the solution by selecting elements with high marginal gain relative to the structural cost imposed by the $p$-matchoid constraints. This recursive mechanism forms the backbone of our dynamic algorithm.

At each level, the algorithm maintains a candidate set and a partial solution. An element is promoted to the next level only if its marginal contribution outweighs the cost of maintaining feasibility across the constituent matroids. Within each level, one element is sampled from the surviving candidates and added to the solution, while conflicting elements are removed. This yields a layered recursive structure culminating in the final solution.

Our dynamic algorithm is inspired by the streaming algorithm of Feldman, Karbasi, and Kazemi Feldman et al. [2018], but technical modifications are required to support efficient updates in a fully dynamic setting. Our key idea is to simulate the behavior of the offline algorithm on an evolving ground set, while ensuring that the amortized update cost remains low.

Our algorithm incorporates two randomized procedures. The first is an initial sampling phase that reduces the input size and transforms a non-monotone problem into a more tractable monotone-like instance. The second is used during updates to guard against adversarially ordered insertions and deletions. When an element is inserted, it is retained with some probability. If retained, it is evaluated for inclusion in the current solution, and affected levels may be reconstructed. Similarly, deletions are handled by checking whether the element contributed to the current solution and selectively rebuilding the necessary components.

This design ensures that each update has low expected cost while maintaining near-optimal performance guarantees. A complete description of the algorithm appears in Section 4.

### 1.1.2 Comparison with prior dynamic and streaming algorithms

Several dynamic and streaming algorithms exist for submodular maximization under various constraints such as cardinality, matroid, and $p$-matchoid constraints. However, our algorithm differs in several key aspects that we summarize below:

**Streaming vs. dynamic models.** The algorithm by Feldman et al. Feldman et al. [2018] is specifically designed for the insertion-only streaming setting, and it cannot naturally handle deletions. In the streaming model, once an element is deleted, we must reassess all previously seen elements to find a replacement—this typically requires $\Omega(n)$ oracle queries. In contrast, the dynamic model demands efficient handling of both insertions and deletions. Applying the streaming algorithm in a dynamic setting would require reconstructing the solution from scratch after every deletion, resulting in linear query complexity per update, which is computationally prohibitive.

**Matroid vs. $p$-matchoid constraints.** $p$-matchoid constraints are a generalization of matroid constraints, which makes adapting existing algorithms more challenging. For example, the algorithm of Banihashem et al. Banihashem et al. [2024] depends on a monotonicity property essential for performing binary search over levels. This property holds in matroids but not in general $p$-matchoids. Hence, their approach cannot be extended to $p$-matchoids.

To illustrate the failure of monotonicity in $p$-matchoids, consider the following counterexample. Let $\mathcal{V} = \{1, 2, 3\}$ and define a 2-matchoid composed of two matroids. The first has ground set $\{1, 2\}$ and allows only one element, i.e., $\mathcal{I}_1 = \{\emptyset, \{1\}, \{2\}\}$. The second has ground set $\{1, 3\}$ with $\mathcal{I}_2 = \{\emptyset, \{1\}, \{3\}\}$. Thus, the independent sets of the matchoid are $\mathcal{I} = \{\emptyset, \{1\}, \{2\}, \{3\}, \{2, 3\}\}$.

Now consider an additive submodular function with $f(\{3\}) < f(\{1\}) < f(\{2\})/10$. Suppose $A_1 = \{1\}$ is selected at level 1. At level 2, adding element 2 requires removing 1 (due to matroid 1), giving $A_2 = \{2\}$. Since 1 is now absent, we can include element 3 at level 3, resulting in $A_3 = \{2, 3\}$. Thus, element 3 can be added at level 3 but not at level 1 due to conflict with 1, violating monotonicity assumptions. This shows the binary search technique used for matroids fails for $p$-matchoids.

**Monotone vs. non-monotone functions.** Our algorithm also handles non-monotone submodular functions, unlike the method of Monemizadeh Monemizadeh [2020], which is designed for monotone functions under cardinality constraints. Monemizadeh's algorithm samples $O(\epsilon^{-2} \log n)$ elements to ensure a sufficiently good one is selected, whereas we require only a single sample due to our algorithmic structure and more refined analysis.

**Cardinality vs. $p$-matchoid constraints.** The algorithm by Banihashem et al. Banihashem et al. [2023] is specific to cardinality constraints and achieves an $(8 + \epsilon)$-approximation for non-monotone submodular functions. Since cardinality is a special case of $p$-matchoid constraints, our algorithm generalizes their result while improving the approximation factor to $(5.82 + \epsilon)$.

## 2 Related work

Submodular maximization under a cardinality constraint $k$ in the dynamic model was first studied at NeurIPS'20 by Lattanzi, Mitrovic, Norouzi-Fard, Tarnawski, and Zadimoghaddam Lattanzi et al. [2020], and independently by Monemizadeh Monemizadeh [2020]. Both papers demonstrated how to maintain a $(2 + \epsilon)$-approximate solution for this problem while having fast query time. The amortized query complexity of the dynamic algorithm presented in Lattanzi et al. [2020] is $O(\epsilon^{-11} \log^6(k) \log^2(n))$ in expectation. And the algorithm presented in Monemizadeh [2020] has an expected amortized $O(\epsilon^{-3} k^2 \log^5(n))$ query complexity. Later, Chen and Peng Chen and Peng [2022] showed that developing a $c$-approximation dynamic algorithm for any $c < 2$ requires an amortized query complexity polynomial in the size of the ground set $V$.

Recently, at ICML 2023, Duetting, Fusco, Lattanzi, Norouzi-Fard, and Zadimoghaddam Duetting et al. [2023] studied monotone submodular maximization under a matroid constraint in the dynamic model and proposed a dynamic $(4 + \epsilon)$-approximation algorithm with an amortized expected query complexity of $O(\frac{k^2}{\epsilon} \log(k) \log^2(n) \log^3(\frac{k}{\epsilon}))$. Simultaneously, Banihashem, Biabani, Goudarzi, Hajiaghayi, Jabbarzade, and Monemizadeh Banihashem et al. [2024] tackled the same problem with the same approximation guarantee but with improved query complexity. Specifically, their algorithm achieves a worst-case expected query complexity of $O(k \log(k) \log^3(\frac{k}{\epsilon}))$. Additionally, the authors in Banihashem et al. [2024] improved the query complexity of the dynamic algorithm proposed in Monemizadeh [2020] for the cardinality constraint to an expected $O(k\epsilon^{-1} \log^2(k))$.

All these results are for monotone submodular maximization problems. The only result for non-monotone submodular maximization within the dynamic setting is a recent result presented at NeurIPS'23 by Banihashem, Biabani, Goudarzi, Hajiaghayi, Jabbarzade, and Monemizadeh Banihashem et al. [2023] who presented a dynamic algorithm for non-monotone submodular maximization under the cardinality constraint $k$. This dynamic algorithm maintains a $(8 + \varepsilon)$-approximate solution for cardinality constraint while having an expected amortized $O(\varepsilon^{-2}k^2 \log^3(k))$ oracle queries per update.

Submodular maximization problems have also been studied in the streaming model. Badanidiyuru, Mirzasoleiman, Karbasi, and Krause Badanidiyuru et al. [2014] investigated monotone submodular maximization under a cardinality constraint $k$ and proposed an insertion-only streaming algorithm with a $(2 + \epsilon)$-approximation guarantee. Chekuri, Gupta, and Quanrud Chekuri et al. [2015] presented a $O(p)$-approximation single-pass streaming algorithm for maximizing non-monotone submodular functions subject to $p$-matchoid constraints using $O(k \log k)$ space. Feldman, Karbasi, and Kazemi Feldman et al. [2018] and Mirzasoleiman, Jegelka, and Krause Mirzasoleiman et al. [2018] subsequently developed streaming algorithms with improved approximation guarantees for maximizing non-monotone functions under $p$-matchoid constraints. Specifically, the streaming algorithm in Feldman et al. [2018], which was an inspiration for our dynamic algorithm has the same approximation guarantee of $= 4p + 2 - o(1)$ as our algorithm, and uses $O(k)$ space and $O(km/p)$ query calls per element.

## 3   Models and definitions

Let $\mathcal{V}$ denote a ground set. A function $f : 2^{\mathcal{V}} \to \mathbb{R}^{\geq 0}$ is termed *submodular* if it satisfies the inequality $f(A \cup \{e\}) - f(A) \geq f(B \cup \{e\}) - f(B)$ for all $A \subseteq B \subseteq \mathcal{V}$ and $e \notin B$. For a subset $A \subseteq \mathcal{V}$ and an element $e \in \mathcal{V}$, the difference $\Delta(e|A) = f(A + e) - f(A)$[2] is commonly referred to as the *marginal gain* of adding $e$ to $A$. If $f$ satisfies $f(A \cup e) \geq f(A)$ for all choices of $A$ and $e$, meaning that all marginal gains are non-negative, we refer to $f$ as *monotone*. Conversely, we refer to $f$ as *non-monotone* if the marginal gain of adding an element to a subset is not necessarily non-negative; that is, there could be a subset $A$ and an element $e \notin A$ such that $f(A \cup e) < f(A)$.

**Matroids and $p$-matchoids.**   A *matroid* $\mathcal{M}(\mathcal{V}, \mathcal{I})$ is defined by a *ground set* $\mathcal{V}$ and a nonempty downward-closed set system $\mathcal{I} \subseteq 2^{\mathcal{V}}$, where $\mathcal{I}$ consists of independent sets. It satisfies the *exchange axiom*: for any pair of independent sets $A, B \in \mathcal{I}$ with $|A| < |B|$, there exists an element $x \in B \setminus A$ such that $A \cup \{x\} \in \mathcal{I}$. Any subset of the ground set $\mathcal{V}$ that is not independent is labeled as *dependent*. A maximal independent set, which becomes dependent upon the addition of any other element, is known as a *basis* for the matroid $\mathcal{M}(\mathcal{V}, \mathcal{I})$. Conversely, a *circuit* in a matroid $\mathcal{M}(\mathcal{V}, \mathcal{I})$ is a minimal dependent subset of $\mathcal{V}$, meaning a dependent set whose proper subsets are all independent. Given a subset $A$ of $V$, the rank of $A$, denoted as *rank(A)*, represents the maximum cardinality of an independent subset within $A$.

An *$p$-matchoid* $\mathcal{M}(\mathcal{V}, \mathcal{I})$ consists of $m$ matroids $\mathcal{M}_1(\mathcal{V}_1, \mathcal{I}_1), \mathcal{M}_2(\mathcal{V}_2, \mathcal{I}_2), \ldots, \mathcal{M}_m(\mathcal{V}_m, \mathcal{I}_m)$ such that each element in the ground set $\mathcal{V}$ appears in the ground sets of at most $p$ of these matroids. The set $\mathcal{I}$ of independent sets is defined as the collection of all subsets of $\mathcal{V}$ whose projection onto any ground set $\mathcal{V}_i$ is an independent set in $\mathcal{I}_i$. In other words, $\mathcal{I} = \left\{ S \subseteq 2^{\mathcal{V}} : \forall_{i=1}^{m} S \cap \mathcal{V}_i \in \mathcal{I}_i \right\}$.

A classic example is the 2-matchoid representing the classical matching problem. In this case, given an unweighted graph $G(V, E)$, a set $M \subseteq E$ is a matching of $G$ if and only if every vertex $u \in V$ is incident to at most one edge of $M$. We can define a 2-matchoid $\mathcal{M}(\mathcal{V}, \mathcal{I})$ for $G$ with ground set $\mathcal{V} = E$ as the edge set of $G$ and independent set $\mathcal{I}$ representing all valid matchings in $G$. Specifically, for each vertex $v$, a matroid $\mathcal{M}_v(\mathcal{V}_v, \mathcal{I}_v)$ is defined, where $\mathcal{V}_v$ represents the edges in $\mathcal{V}$ incident to $v$. An edge set $X \subseteq \mathcal{V}$ is independent in $\mathcal{M}_v(\mathcal{V}_v, \mathcal{I}_v)$ if $|X \cap \mathcal{V}_v| \leq 1$. Thus, any edge set $X \subseteq \mathcal{V}$ is an independent set in the 2-matchoid $\mathcal{M}(\mathcal{V}, \mathcal{I})$ (i.e., $X \in \mathcal{I}$) if and only if $X$ is a valid matching in $G$.

**Definition 3.1** (*$p$-matchoid submodular maximization*).  Consider a submodular function $f : 2^{\mathcal{V}} \to \mathbb{R}^{\geq 0}$ (not necessarily monotone). The goal in *$p$-matchoid submodular maximization* or submodular maximization subject to the $p$-matchoid constraint $\mathcal{M}(\mathcal{V}, \mathcal{I})$ is to find an independent set $I^* \in \mathcal{I}$ that maximizes $f(I^*)$. In other words, if we denote the maximum submodular value of an independent

---

[2]For a set $A$ and an element $e$, we may represent the union of sets $A$ and $\{e\}$ as $A + e$ or $A \cup e$ for convenience. Similarly, when considering a set $A$ with an element $e \in A$, we may denote $A \setminus \{e\}$ as $A - e$ or $A \setminus e$.

set in $\mathcal{I}$ as $OPT = \max_{I \in \mathcal{I}} f(I)$, then $I^* \in \mathcal{I}$ is an independent set that achieves the optimal value $f(I^*) = OPT$.

**Oracle queries.** We assume that the access to matchoid $\mathcal{M}(\mathcal{V}, \mathcal{I})$ is through an oracle that answers the following types of queries.

- **Submodular value oracle:** The oracle provides access to a submodular function $f : 2^{\mathcal{V}} \rightarrow \mathbb{R}^{\geq 0}$, allowing retrieval of the value $f(A)$ for any subset $A \subset \mathcal{V}$. In this query access model, computing the marginal gain $f(A \cup \{e\}) - f(A)$ is achieved through two queries: $f(A \cup \{e\})$ and $f(A)$, where $A \subseteq \mathcal{V}$ and $e \in \mathcal{V}$.

- **Matroid independence oracle:** The oracle allows access to a matroid $\mathcal{M}_i(\mathcal{V}_i, \mathcal{I}_i)$ of a $p$-matchoid $\mathcal{M}(\mathcal{V}, \mathcal{I})$. For any subset $S \subseteq \mathcal{V}$, the oracle answers whether $S$ is an independent set in matroid $\mathcal{M}_i(\mathcal{V}_i, \mathcal{I}_i)$ or if it is dependent. In other words, it determines if $S \in \mathcal{I}_i$.

We assess the *time complexity* of an algorithm that (approximately) solves $p$-matchoid submodular maximization in terms of its *query complexity*, defined as the number of queries made to either the submodular value oracle for $f$ or the matroid independence oracle for $\mathcal{I}$.

**Dynamic Model.** Consider a sequence $\mathcal{S}$ comprising insertions and deletions of elements from an underlying ground set $\mathcal{V}$. Let $\mathcal{S}_t$ denote the sequence of the first $t$ updates (insertions or deletions) from $\mathcal{S}$. The term *time $t$* refers to the moment after the first $t$ updates of the sequence $\mathcal{S}$ have been executed. We define $V_t$ as the set of elements that have been inserted until time $t$ but have not been deleted since their latest insertion.

In *dynamic $p$-matchoid submodular maximization*, our goal is to have an approximate solution of $OPT_t = \max_{I_t \subseteq V_t : I_t \in \mathcal{I}} f(I_t)$ at any time $t$. We use the term *dynamic $\alpha$-approximate algorithm* to refer to such an algorithm whose output solution at every timestep $t$ is guaranteed to have a function value that is at least $\alpha^{-1} \cdot OPT_t$.

The performance of dynamic algorithms is evaluated based on their *update time*, which measures the computational effort required to maintain a feasible solution between two consecutive updates. Accordingly, the *query complexity* of a dynamic $\alpha$-approximate algorithm for $p$-matchoid submodular maximization denotes the number of oracle queries (value or independence) required to compute an independent set whose submodular value is an $\alpha$-approximation of $OPT_t$, given all computations performed up to time $t - 1$.

Our dynamic algorithm operates within the *oblivious adversarial model* Carter and Wegman [1977]. In this model, the adversary determines the elements in the set $V$ and their arrival order. However, the adversary remains unaware of the random bits used in the algorithm and, therefore, cannot adaptively choose updates in response to the algorithm's randomly guided choices. It is assumed that the adversary prepares the complete input sequence (of insertions and deletions) before the algorithm starts running.

# 4 Offline and Dynamic algorithms for $p$-matchoid

First, we explain our offline method provided in Algorithm 1 that given a ground set $V$, computes an approximate solution for submodular maximization subject to $p$-matchoid constraints. This algorithm is fully executed by invoking the INIT($V$) procedure, which solves the problem in the offline scenario or acts as a pre-processing step in the dynamic scenario, in which case $V$ may be either empty or not. Then, we show how we handle the insertion and deletion of elements in Algorithm 2. Note that throughout the paper, we assume access to a parameter MAX that approximates the maximum submodular value up to a factor of 2; formally, $\max_{v \in \mathcal{V}} f(v) \leq \text{MAX} \leq 2 \max_{v \in \mathcal{V}} f(v)$. This is a standard assumption, which we show how to lift by maintaining parallel runs in Appendix B.

## 4.1 Offline algorithm

In the beginning, our offline algorithm employs a random sampling procedure named RATESAMPLING that intuitively transforms a non-monotone submodular maximization instance into a monotone one. Once the initial sampling is complete, the algorithm uses a recursive procedure to refine a solution from the remaining elements at different levels. At any level $i$, we maintain two sets $A_i$ and $B_i$. The set $B_i$ is the set of every element $e \in B_{i-1}$ whose marginal gain in case of addition to $A_{i-1}$ is large enough

---
**Algorithm 1** MATCHOIDCONSTRUCTION
---

**Procedure** INIT($V$):
1: $A_0 \leftarrow \emptyset$, and $B_0 \leftarrow$ RATESAMPLING($V$)
2: Return RECURSION($A_0, B_0$)

**Procedure** RATESAMPLING($V$):
1: Sample each element $e \in V$ i.i.d with probability $q = (p + \sqrt{p^2 + p} + 1)^{-1}$
2: Return sampled set $B_0$ in Step 1

**Procedure** RECURSION($A_{i-1}, B_{i-1}$):
1: $B_i \leftarrow$ FILTERING($A_{i-1}, B_{i-1}$)
2: **if** $|B_i| > 0$ **then**
3:    Sample an element $e_i \in B_i$ uniformly at random
4:    $A_i \leftarrow$ EXTENSION($A_{i-1}, e_i$)
5:    Return RECURSION($A_i, B_i$)
6: Let $i^* \leftarrow i - 1$
7: Return $A_{i^*}$

**Procedure** FILTERING($A_{i-1}, B_{i-1}$):
1: $B_i \leftarrow \emptyset$
2: **for** every element $e \in B_{i-1}$ **do**
3:    $U_i(e) \leftarrow$ FINDSWAPS($A_{i-1}, e$)
4:    store $U_i(e)$ for element $e$ as it may be used later in Procedure EXTENSION
5:    **if** $U_i(e) \neq$ FAIL **then**
6:       $B_i \leftarrow B_i \cup e$
7: Return $B_i$

**Procedure** EXTENSION($A_{i-1}, e_i$):
1: Set weight $w(e_i) \leftarrow \Delta(e_i | A_{i-1})$
2: $A_i \leftarrow A_{i-1} - U_i(e_i) + e_i$
3: Return $A_i$

**Procedure** FINDSWAPS($A_{i-1}, e$):
1: $U_i(e) \leftarrow$ DEPENDENCYDETECTION($A_{i-1}, e$)
2: $c \leftarrow \sqrt{1 + \frac{1}{p}}$
3: $\tau \leftarrow \max\left\{(1 + c) \cdot w(U_i(e)), \frac{\varepsilon \text{MAX}}{2k}\right\}$
4: **if** $\Delta(e | A_{i-1}) \geq \tau$ **then**
5:    Return $U_i(e)$
6: **else**
7:    Return FAIL

**Procedure** DEPENDENCYDETECTION($A_{i-1}, e$):

1: $U_i(e) \leftarrow \emptyset$
2: **for** $j = 1$ to $m$ **do**
3:    **if** $(A_{i-1} + e) \cap \mathcal{V}_j \notin \mathcal{I}_j$ **then**
4:       $X_j \leftarrow \{e' \in A_{i-1} | (A_{i-1} - e' + e) \cap \mathcal{V}_j \in \mathcal{I}_j\}$
5:       $x_j \leftarrow \arg\min_{e' \in X_j} w(e')$
6:       $U_i(e) \leftarrow U_i(e) + x_j$
7: Return $U_i(e)$

---

to offset the loss of the elements we would need to delete from $A_{i-1}$ to preserve its independence, and the set $A_i$ is the solution computed at the first $i$ levels. In the end, we denote the number of the last constructed level by $i^*$, and $A_{i^*}$ would be our final solution. This recursive procedure relies on a filtering process and a second sampling process designed to outmaneuver the adversary.

**Initial sampling** The first sampling process samples elements of $V$ with probability $q = (p + \sqrt{p^2 + p} + 1)^{-1} \in O(p^{-1})$ and discards elements that are not sampled. This simple sampling process is well-established Fahrbach et al. [2019], Mirzasoleiman et al. [2018], Feldman et al. [2018] as a technique to transform an instance of non-monotone submodular maximization into a monotone one. It even speeds up the algorithm as it reduces the number of elements to process.

---
**Algorithm 2** UPDATEOPERATIONS
---

**Procedure** INSERT($e$):
1: With probability $1 - q$ Ignore $e$ and return
2: $B_0 \leftarrow B_0 + e$
3: **for** $i \leftarrow 1$ to $i^*$ **do**
4:    $U_i(e) \leftarrow$ FINDSWAPS($A_{i-1}, e$)
5:    **if** $U_i(e) = FAIL$ **then**
6:       Return
7:    $B_i \leftarrow B_i + e$
8:    With probability $r_i = \frac{1}{|B_i|}$:
      ①   $e_i \leftarrow e$
      ②   $A_i \leftarrow$ EXTENSION($A_{i-1}, e_i$)
      ③   Return RECURSION($A_i, B_i$)

**Procedure** DELETE($e$):
1: **for** $i \leftarrow 0$ to $i^*$ **do**
2:    **if** $e \notin B_i$ **then**
3:       Return
4:    $B_i \leftarrow B_i - e$
5:    **if** $e_i = e$ **then**
6:       **if** $|B_i| > 0$ **then**
7:          Sample an element $e_i \in B_i$ uniformly at random
8:          $A_i \leftarrow$ EXTENSION($A_{i-1}, e_i$)
9:          Return RECURSION($A_i, B_i$)
10:       Let $i^* \leftarrow i - 1$
11:       Return $A_{i^*}$

---

**Filtering.** The inputs to the filtering procedure at an arbitrary level $i$ are the sets $A_{i-1}$ and $B_{i-1}$. In this procedure, for each $e \in B_{i-1}$ we decide whether to keep or filter this element at this level based on its suitability for addition to $A_{i-1}$. Note that if $e$ is later selected for addition to $A_{i-1}$, we fix its weight as $w(e) = \Delta(e \mid A_{i-1})$, where $\Delta(e \mid A) := f(A \cup \{e\}) - f(A)$ denotes the marginal gain of $e$ with respect to $A$.

Adding $e$ to $A_{i-1}$ may violate its independency. To address this, we compute a set $U_i$ of elements that would need to be removed to make it independent again. Formally, we start with an empty set $U_i$, then for each matroid $\mathcal{M}_j(\mathcal{V}_j, \mathcal{I}_j)$ of $p$-matchoid $\mathcal{M}(\mathcal{V}, \mathcal{I})$ for which $(A_{i-1} + e) \cap \mathcal{V}_j \notin \mathcal{I}_j$, we identify the element $e' \in A_{i-1}$ with minimum weight that $(A_{i-1} + e - e') \cap \mathcal{V}_j \in \mathcal{I}_j$ and add $e'$ to $U_i(e)$.

Once $U_i(e)$ is computed, we consider two cases: If the marginal gain of adding $e$ to $A_{i-1}$ is significant enough to offset deleting $U_i(e)$, specifically if $\Delta(e|A_{i-1}) \geq (1 + \sqrt{1 + \frac{1}{p}}) \cdot \sum_{e' \in U_i(e)} w(e')$, then $e$ will be added to a survivor set $B_i$, which was initialized as an empty set at the beginning of this level. If this condition is not met, $e$ gets filtered out.

**Sampling and EXTENSION.** At the end of the filtering step in level $i$, set $B_i$ is the set of survivors. We sample one of the survivors, say $e_i \in B_i$, and obtain the new solution set $A_i = A_{i-1} + e_i - U_i(e_i)$. In addition, we set the weight of sampled element $e_i$ to $w(e_i) = \Delta(e_i|A_{i-1})$.

Our sampling strategy is similar to that of Monemizadeh Monemizadeh [2020] who used a similar method for dynamic (monotone) submodular maximization under a cardinality constraint. However, in that work, he samples $O(\epsilon^{-2} \log(n))$ elements to guarantee that one of the sampled elements has a marginal gain above a predetermined threshold. In contrast, our sampling strategy requires sampling just one element, and our filtering guarantees that the marginal gain of this element is sufficiently large.

## 4.2 Insertion and deletion algorithms

Upon the insertion of an element $e$, we ignore $e$ with probability $1 - q$ and terminate the insertion algorithm. Otherwise (i.e., with probability $q$), we iterate through levels $i \in \{1, \dots, i^*\}$ and check if we can add $e$ to the survivor set $B_i$. If this is the case, we then reconstruct levels $i, \dots, i^*$ with probability $\frac{1}{|B_i|}$.

For deleting an element $e$, we iterate through levels $\{1, \dots, i^*\}$. At each level $i$, we check if $e$ is the element that has been sampled at level $i$. If this is the case, similar to the insertion algorithm, we reconstruct levels $i, \dots, i^*$ in a recursive manner.

## 5 Analysis

In this section, we provide a sketch of our analysis including some notations and definitions. The detailed proofs are provided in Appendix A.

Survivors are elements of a set $B_{i-1}$ that are included in set $B_i$. Formally, we define them as follows:

**Definition 5.1** (Survivor). Let $1 \leq i \leq i^*$ be a level. We call an element $e$, a survivor for level $i$ if $\Delta(e|A_{i-1}) \geq (1 + \sqrt{1 + \frac{1}{p}}) \cdot \sum_{e' \in U_i(e)} w(e')$.

In this definition, $U_i(e)$ denotes the set of dependent elements that must get removed from the independent set $A_{i-1}$ so that $e$ can be added to it. More precisely, for every matroid $\mathcal{M}_j(\mathcal{V}_j, \mathcal{I}_j)$ in the matchoid $\mathcal{M}(\mathcal{V}, \mathcal{I})$, if $(A_{i-1} + e) \cap \mathcal{V}_j \notin \mathcal{I}_j$, we define $X_j = \{e' \in A_{i-1} \mid ((A_{i-1} - e' + e) \cap \mathcal{V}_j) \in \mathcal{I}_j\}$, the set of elements whose removal would restore independence in matroid $\mathcal{M}_j(\mathcal{V}_j, \mathcal{I}_j)$ after the addition of $e$. We then select the element $x_j = \arg\min_{e' \in X_j} w(e')$ with the smallest weight among those in $X_j$ and add it to $U_i(e)$.

In the analysis of our randomized algorithm, we denote the random variable itself as $\mathbf{x}$ for any variable $x$, and $x$ represents its value during execution. The key random variables used in our analysis are as follows:

- We denote $\mathbf{e}_i$ as the random variable corresponding to the sampled element $e_i$ at level $i$.

- $\mathbf{B}_i$ represents the random variable corresponding to the set $B_i$.

- The random variable $\mathbf{i}^*$ corresponds to $i^*$, the index of the last non-empty level created.
- We define $\mathbf{C}_i = (\mathbf{e}_1, \ldots, \mathbf{e}_{i-1}, \mathbf{B}_0, \ldots, \mathbf{B}_i)$ as the random variable corresponding to configuration $C_i = (e_1, \ldots, e_{i-1}, B_0, \ldots, B_i)$ up to level $i$.

When observing an update at time $t$, it is crucial to differentiate between random variables $\mathbf{e}_i$, $\mathbf{B}_i$, $\mathbf{i}^*$, and $\mathbf{C}_i$ and their respective values before and after the update. To this end, we employ the notations $\mathbf{Y}^-$ and $Y^-$ to represent a random variable and its value before time $t$. We continue to use $\mathbf{Y}$ and $Y$ to denote them at the current time after the execution of the update.

We divide the analysis of the dynamic algorithm algorithm into several steps. We first define a set of invariants and we show that we can maintain them in the course of the dynamic algorithm. Having these invariants in hand, we prove the approximation guarantee and the query complexity of the algorithm.

**Step 1: Invariants.**  Initially, we introduce the following set of invariants.

1. *Survivor:* $B_i = \{e \in B_{i-1} : \Delta(e|A_{i-1}) \geq (1 + \sqrt{1 + \frac{1}{p}}) \cdot \sum_{e' \in U_i(e)} w(e')\}$ for all $1 \leq i \leq i^* + 1$.

2. *Starter:* $B_0 = \textsc{RateSampling}(V)$ and $A_0 = \emptyset$

3. *Weight:* For $1 \leq i \leq i^*$, $e_i \in B_i$ and $w(e_i) = \Delta(e_i|A_{i-1})$

4. *Independent:* For $1 \leq i \leq i^*$, $A_i = A_{i-1} + e_i - U_i(e_i)$

5. *Terminator:* $B_{i^*+1} = \emptyset$

6. *Uniform invariant:* $\mathbb{P}\left[\mathbf{e}_i = e | \mathbf{i}^* \geq i \text{ and } \mathbf{C}_i = C_i\right] = \frac{1}{|B_i|} \cdot \mathbb{1}\left[e \in B_i\right]$.

**Step 2: Maintenance of invariants.**  Next, we demonstrate the validity of the invariants at the end of the execution of \textsc{MatchoidConstruction}. We also show that these invariants remain preserved following each insertion and deletion operation. Specifically, we establish the following lemmas.

**Lemma 5.2.**  *At the end of Algorithm* \textsc{MatchoidConstruction} *all invariants hold.*

**Lemma 5.3.**  *If before the insertion or deletion of an element $e$, the invariants hold, then they also hold after the execution of* \textsc{Insert}$(e)$ *and* \textsc{Delete}$(e)$, *respectively.*

**Step 3: Query complexity.**  Subsequently, we establish bounds on the expected worst-case query complexity of both insertion and deletion operations. Formally, we prove the following result.

**Theorem 5.4.**  *The expected query complexity of each insert/delete for all runs is $O(\varepsilon^{-3} p k^4 \log^2(k))$.*

**Step 4: Approximation guarantee.**  Finally, we demonstrate that, assuming the invariants hold, an independent set $A_{i^*} \in \mathcal{I}$ of the matchoid $\mathcal{M}(\mathcal{V}, \mathcal{I})$ can be reported, with a submodular value approximating the optimal solution by $(2p + 2\sqrt{p(p+1)} + 1 + \varepsilon)$.

**Theorem 5.5.**  *Suppose that the invariants hold in every run of* \textsc{UpdateOperations}. *Let $A_{i^*}$ be the independent set that Algorithm* \textsc{UpdateOperations} *returns. Then, the set $A_{i^*}$ satisfies $(2p + 2\sqrt{p(p+1)} + 1 + \varepsilon) \cdot f(A_{i^*}) \geq OPT$, where $OPT = \max_{I^* \in \mathcal{I}} f(I^*)$.*

## Acknowledgements

The work is partially supported by DARPA QuICC, ONR MURI 2024 award on Algorithms, Learning, and Game Theory, Army-Research Laboratory (ARL) grant W911NF2410052, NSF AF:Small grants 2218678, 2114269, 2347322, and Royal Society grant IES\R2\222170.

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

# A  Omitted proofs

We begin proving the main theorem of our paper by first introducing basic probabilistic terminology.

**Conditional expectation.**  For a function $x$ and a set $A$, we denote $x[A]$ as the function $x$ restricted to the domain $A$. For an event $E$, we define $\mathbb{1}[E]$ as the *indicator function* of $E$, i.e., $\mathbb{1}[E]$ is set to one if $E$ holds and zero otherwise. When dealing with random variables and their values, we use bold and non-bold letters, respectively. For instance, a random variable is denoted as $\mathbf{X}$, and its value is represented by $X$. We use the notations $\mathbb{P}[\mathbf{X}]$ and $\mathbb{E}[\mathbf{X}]$ for the probability and expectation of a random variable $\mathbf{X}$. For events $A$ and $B$, the notation $\mathbb{P}[A|B]$ denotes "*the conditional probability of A given B*" or "*the probability of A under the condition B*". For an event $A$ with nonzero probability and a discrete random variable $\mathbf{X}$, we denote by $\mathbb{E}[\mathbf{X}|A]$ the conditional expectation of $X$ given $A$, i.e., $\mathbb{E}[\mathbf{X}|A] = \sum_x x \cdot \mathbb{P}[\mathbf{X} = x|A]$. Similarly, for discrete random variables $\mathbf{X}$ and $\mathbf{Y}$, the conditional expectation of $\mathbf{X}$ given $\mathbf{Y}$ is denoted by $\mathbb{E}[\mathbf{X}|\mathbf{Y} = y]$.

## A.1  Maintenance of survivor invariant

We first prove that the survivor invariant holds at the end of the algorithm MATCHOIDCONSTRUCTION. Later, we show that it holds after the execution of INSERT($e$) and DELETE($e$) provided it was upheld before the insertion or deletion of an element $e$. We employ a similar methodology to demonstrate the preservation of the remaining invariants. In particular, we first prove them for the algorithm MATCHOIDCONSTRUCTIONand then for INSERT($e$) and DELETE($e$).

**Survivor invariant for MATCHOIDCONSTRUCTION($\mathcal{V}$):**  We inductively prove that survivor invariant holds for any element $e$ and any level $i < i^*$. For the base case $i = 0$, if any arbitrary element $e \in \mathcal{V}$ is sampled with probability $q$, element $e$ will be in $B_0$. Otherwise, it will not be added to $B_0$.

Now, let us consider an arbitrary element $e$ that is in survivor set $B_{i-1}$ of a level $i - 1$ where $1 \leq i < i^*$. Recall that $U_i(e)$ is the set of dependent elements that needs to be deleted from independent set $A_{i-1}$ so that $e$ can be added to it. In the algorithm FILTERING, we check if $\Delta(e|A_{i-1}) \geq (1 + \sqrt{1 + \frac{1}{p}}) \cdot \sum_{e' \in U_{i-1}(e)} w(e')$. If that is the case, $e$ is added to $B_i$, otherwise, $e$ is not added to $B_i$. Thus, the survivor invariant holds.

**Survivor invariant for INSERT($e$):**  We first give a few useful facts and for that, define variables $r$ and $s$ as follows.

- ✫ Let $s$ be the largest $i \in [0, i^{*-} + 1]$ such that $e$ is added to $B_i^-$.

- ✫ If there is a level $i \in [i^{*-}]$ in which $e_i = e$, then we let $r$ be $i$. Otherwise, we let $r$ be $i^{*-} + 1$. We consider two cases.

    - ☞ Case 1 happens if $r \leq i^{*-}$. Then, we know that Subroutines EXTENSION and RECURSION have been invoked. In addition, we have $e_r = e$ and $s = r$.

    - ☞ Case 2 occurs if $r = i^{*-} + 1$ which means Subroutines EXTENSION and RECURSION have never been invoked for INSERT($e$). Thus, the last level $i^*$ after inserting $e$ is the same last level $i^{*-}$ before inserting $e$. That is, $i^* = i^{*-}$. Note that, we must have $s \neq i^{*-} + 1$, since otherwise, we have $|B_{i^*+1}^-| = 0$ and $|B_{i^*+1}| = 1$. Therefore, we have invoked Subroutines EXTENSION and RECURSION for this level what means $i^* > i^{*-}$ which is not the case. Thus, $s < r = i^{*-} + 1$.

We handle INSERT($v$). Therefore, for any $i < r$, we have not made any change in variables $e_i^-$, $w(e_i)$, or $A_i^-$ because we have not invoked Subroutines EXTENSION and RECURSION for those levels upon insertion of $e$. Hence, the following facts are correct.
*Fact* A.1.  For any $i \in [1, r)$ we have: ❶ $e_i = e_i^-$  ❷ $w(e_i) = w^-(e_i)$  ❸ $A_i = A_i^-$. For $i = 0$, we have $A_i = A_i^-$.

By the definition of $s$, we add $e$ to the set $B_i^-$, for each $i \in [0, s]$ and we have $s \leq r$. In addition, by invoking Subroutines EXTENSION and RECURSION for level $r + 1$, nothing happens to the variables in levels that are less than $r + 1$. Thus, we obtain the following fact.
*Fact* A.2.  For any $i \in [1, s]$, we have $B_i = B_i^- + e$.
*Fact* A.3.  For any $i \in [s + 1, r]$, we have $B_i = B_i^-$.

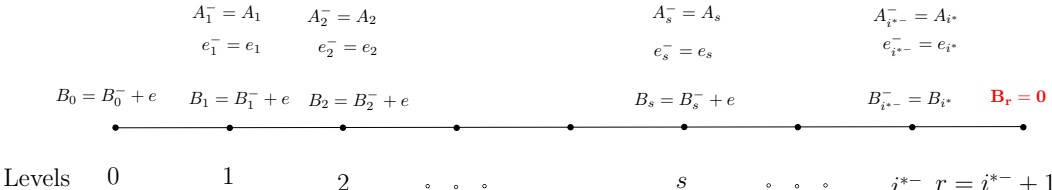

Case 1 of inserting an element $e$

$A_1^- = A_1$  $A_2^- = A_2$  $A_s^- = A_s$  $A_{i^*-}^- = A_{i^*}$
$e_1^- = e_1$  $e_2^- = e_2$  $e_s^- = e_s$  $e_{i^*-}^- = e_{i^*}$

$B_0 = B_0^- + e$  $B_1 = B_1^- + e$  $B_2 = B_2^- + e$  $B_s = B_s^- + e$  $B_{i^*-}^- = B_{i^*}$  $\mathbf{B_r = 0}$

Levels  0  1  2  $\circ$ $\circ$ $\circ$  $s$  $\circ$ $\circ$ $\circ$  $i^{*-}$  $r = i^{*-} + 1$

Case 2 of inserting an element $e$

Figure 1: Visualisation of two cases upon INSERT($e$).

Next, we show that the survivor invariant holds, i.e., $B_i = \{e \in B_{i-1} : \Delta(e|A_{i-1}) \geq (1 + \sqrt{1 + \frac{1}{p}}) \cdot \sum_{e' \in U_i(e)} w(e')\}$ for $i \in [1, i^*]$.

First, let us consider $i \in [1, s]$. By the assumption, the survivor invariant holds before the the insertion of element $e$ at time $t$. That is, $B_i^- = \{e \in B_{i-1}^- : \Delta(e|A_{i-1}^-) \geq (1 + \sqrt{1 + \frac{1}{p}}) \cdot \sum_{e' \in U_i(e)} w^-(e')\}$ for $i \in [1, s]$. Using Fact A.2, $B_i = B_i^- + v$ holds for all $i \in [1, s]$. Thus,

$$B_i = B_i^- + v = \{e \in B_{i-1}^- : \Delta(e|A_{i-1}^-) \geq (1 + \sqrt{1 + \frac{1}{p}}) \cdot \sum_{e' \in U_i(e)} w^-(e')\} + v$$

$$= \{e \in B_{i-1}^- \cup \{v\} : \Delta(e|A_{i-1}^-) \geq (1 + \sqrt{1 + \frac{1}{p}}) \cdot \sum_{e' \in U_i(e)} w^-(e')\} .$$

According to Fact A.1, for any $i \in [1, s] \subseteq [1, r)$ we have: ❶ $e_i = e_i^-$  ❷ $w(e_i) = w^-(e_i)$  ❸ $A_i = A_i^-$. Therefore, we obtain the following:

$$B_i = B_i^- + v = \{e \in B_{i-1} : \Delta(e|A_{i-1}) \geq (1 + \sqrt{1 + \frac{1}{p}}) \cdot \sum_{e' \in U_i(e)} w(e')\} .$$

Recall that if Case 1 occurs, $r = s$. Since in this case, Subroutines EXTENSION and RECURSION have been invoked, we know that the survivor invariant will be held for all level $[s + 1, i^*]$. However, if Case 2 happens, we need to prove that the survivor invariant is held for all levels $[s + 1, i^*]$.

Thus, we are in Case 2 and we need to prove the survivor invariant for all levels $[s + 1, i^*]$. We have $r = i^{*-} + 1 = i^* + 1$. According to Fact A.3, for any $i \in [s + 1, i^* + 1]$, we have $B_i = B_i^-$.

$$B_i = B_i^- = \{e \in B_{i-1}^- : \Delta(e|A_{i-1}^-) \geq (1 + \sqrt{1 + \frac{1}{p}}) \cdot \sum_{e' \in U_i(e)} w^-(e')\} .$$

Once again, using Fact A.1, for any $i \in [1, r)$ we have: ❶ $e_i = e_i^-$ ❷ $w(e_i) = w^-(e_i)$ ❸ $A_i = A_i^-$ what means

$$B_i = \{e \in B_{i-1} : \Delta(e|A_{i-1}) \geq (1 + \sqrt{1 + \frac{1}{p}}) \cdot \sum_{e' \in U_i(e)} w(e')\} \ .$$

This essentially means that the survivor invariant holds after INSERT($e$).

**Survivor invariant for DELETE($e$):**   Similar to INSERT($e$), We first define variables $r$ and $s$ as follows.

☆ Let $s$ be the largest $i \in [0, i^{*-} + 1]$ from which $e$ is deleted from $B_i^-$.

☆ If there is a level $i \in [1, i^{*-}]$ in which $e_i = e$, then we let $r$ be $i$. Otherwise, we let $r$ be $i^{*-} + 1$. We consider two cases.

☛ Case 1 happens if $r \leq i^{*-}$. Then, we know that Subroutines EXTENSION and RECURSION have been invoked. In addition, we have $e_r = e$ and $s = r$.

☛ Case 2 occurs if $r = i^{*-} + 1$ which means Subroutines EXTENSION and RECURSION have never been invoked for INSERT($e$). Thus, the last level $i^*$ after deleting $e$ is the same last level $i^{*-}$ before deleting $e$. That is, $i^* = i^{*-}$. Note that, we must have $s < r = i^{*-} + 1$, since otherwise, we have $|B_{i^{*-}+1}^-| = 1$ which cannot be the case.

Our update is DELETE($v$). Thus, in any level $i \in [1, i^*)$, we do not make any change in in these levels other than removing the element $e$ from $B_i^-$ if $i \leq s$.
*Fact* A.4.  For any $i \in [1, r)$ we have: ❶ $e_i = e_i^-$ ❷ $w(e_i) = w^-(e_i)$ ❸ $A_i = A_i^-$. For $i = 0$, we have $A_i = A_i^-$.
*Fact* A.5.  For any $i \in [0, s]$, it holds that $B_i = B_i^- \setminus \{v\}$ and for any $i \in [s + 1, r]$, we have $B_i = B_i^-$.

By the assumption, the survivor invariant holds before the the deletion of element $e$ at time $t$. That is, $B_i^- = \{e \in B_{i-1}^- : \Delta(e|A_{i-1}^-) \geq (1 + \sqrt{1 + \frac{1}{p}}) \cdot \sum_{e' \in U_i(e)} w^-(e')\}$ for $i \in [1, s]$. Thus, using Fact A.5, for $i \in [1, s]$

$$B_i = B_i^- \setminus \{e\} = \{e \in B_{i-1}^- \setminus \{e\} : \Delta(e|A_{i-1}^-) \geq (1 + \sqrt{1 + \frac{1}{p}}) \cdot \sum_{e' \in U_i(e)} w^-(e')\} \ ,$$

and for $i \in [s + 1, r]$,

$$B_i = B_i^- = \{e \in B_{i-1}^- : \Delta(e|A_{i-1}^-) \geq (1 + \sqrt{1 + \frac{1}{p}}) \cdot \sum_{e' \in U_i(e)} w^-(e')\} \ .$$

Since by Fact A.4, for any $i \in [1, r)$ we have ❶ $e_i = e_i^-$ ❷ $w(e_i) = w^-(e_i)$ ❸ $A_i = A_i^-$, we conclude that

$$B_i = B_i^- \setminus \{e\} = \{e \in B_{i-1} : \Delta(e|A_{i-1}) \geq (1 + \sqrt{1 + \frac{1}{p}}) \cdot \sum_{e' \in U_i(e)} w(e')\} \ ,$$

what proves that the survivor invariant holds for $i \in [1, r)$ after the deletion of element $e$. Recall that if $r \leq i^{*-}$, Subroutines EXTENSION and RECURSION have been invoked, therefore, we know that the survivor invariant will be held for all levels $[r, i^*]$.

## A.2   Maintenance of starter invariant

**Starter invariant for INSERT($e$):**   Recall that in Subroutine RATESAMPLING($\mathcal{V}$), we sample each element $e \in \mathcal{V}$ i.i.d with probability $q = (p + \sqrt{p^2 + p} + 1)^{-1}$ and let $B_0$ be the set of elements that are sampled. By our assumption, we have $B_0^- = $ RATESAMPLING($\mathcal{V}^-$) and $A_0^- = 0$.

Upon insertion of an element $e$, we add it to $B_0^-$ with probability $q$. Observe that we always add $e$ to the ground set. Thus, $\mathcal{V} = \mathcal{V}^- + e$. Condition of this event which happens with probability $q$, we have $B_0 = B_0^- + e$. Thus, $B_0 = $ RATESAMPLING($\mathcal{V}^- + e$) = RATESAMPLING($\mathcal{V}$). In addition, using Fact A.1, we have $A_0 = A_0^- = 0$.

**Starter invariant for DELETE($e$):** By our assumption, we have $B_0^- = $ RATESAMPLING($\mathcal{V}^-$) and $A_0^- = 0$. Moreover, $\mathcal{V}_0 = \mathcal{V}_0^- \setminus \{e\}$. If $e \in B_0^-$ (what happens with probability $q$), we then remove it from $B_0^-$. That is, $B_0 = B_0^- \setminus \{e\} = $ RATESAMPLING($\mathcal{V}^- \setminus \{e\}$) = RATESAMPLING($\mathcal{V}$). Using Fact A.4, $A_i = A_i^- = 0$.

This proves that the starter invariant holds after insertion or deletion of an arbitrary element $e$.

## A.3  Maintenance of independent invariant

We assume that the independent invariant holds until time $t$. That is, for any level $1 \leq i \leq i^{*-}$, $A_i^- = A_{i-1}^- + e_i^- - U_i(e_i^-)$. Upon the insertion or deletion of an arbitrary element $e$ at time $t$, we show that $A_i = A_{i-1} + e_i - U_i(e_i)$ for any level $1 \leq i \leq i^*$.

Remarkably, we can establish this invariant for both insertions and deletions simultaneously, without the need for separate consideration of each operation. Indeed, according to Facts A.1 and A.4, upon the insertion or deletion of an arbitrary element $e$ at time $t$, for any $i \in [1, r)$ we have ❶ $e_i = e_i^-$ ❷ $w(e_i) = w^-(e_i)$ ❸ $A_i = A_i^-$ and for $i = 0$, we have $A_i = A_i^-$. Thus, $A_i = A_i^- = A_{i-1}^- + e_i^- - U_i(e_i^-)$.

Now, observe that for any $j \leq i - 1 < r$, we have ❶ $e_j = e_j^-$ ❷ $w(e_j) = w^-(e_j)$ ❸ $A_j = A_j^-$ and for $j = 0$, we have $A_j = A_j^-$. This essentially means that $A_i = A_{i-1} + e_i - U_i(e_i)$ for any level $1 \leq i \leq i^*$. Also, recall that if $r \leq i^{*-}$, Subroutines EXTENSION and RECURSION have been invoked, therefore, we know that the independent invariant will be held for all levels $[r, i^*]$.

## A.4  Maintenance of weight invariant

Next, we show that the weight invariant holds. That is, assuming that for $1 \leq i \leq i^{*-}$, $e_i^- \in B_i^-$ and $w^-(e_i^-) = \Delta(e_i^-|A_{i-1}^-)$. We show that upon the insertion or deletion of an arbitrary element $e$ at time $t$, $e_i \in B_i$ and $w(e_i) = \Delta(e_i|A_{i-1})$ for $1 \leq i \leq i^*$.

**Weight invariant for INSERT($e$):** According to Fact A.2, for any $i \in [1, s]$, we have $B_i = B_i^- + e$ and for any $i \in [s + 1, r]$, we have $B_i = B_i^-$ using Fact A.3. Thus, $B_i^- \subseteq B_i$.

Using Fact A.1, for any $i \in [1, r)$ we have ❶ $e_i = e_i^-$ ❷ $w(e_i) = w^-(e_i)$ ❸ $A_i = A_i^-$. For $i = 0$, we have $A_i = A_i^-$. Therefore, $e_i = e_i^- \in B_i^- \subseteq B_i$ for any $i \in [1, i^*)$. This essentially means that $e_i \in B_i$.

Next we need to show $w(e_i) = \Delta(e_i|A_{i-1}) = f(A_{i-1} + e_i) - f(A_{i-1})$. For any $i \in [1, r)$, we have $w(e_i) = w^-(e_i)$ and $e_i = e_i^-$. Then, $w^-(e_i) = w^-(e_i^-)$. Moreover, the assumption of this lemma implies that $w^-(e_i^-) = \Delta(e_i^-|A_{i-1}^-) = f(A_{i-1}^- + e_i^-) - f(A_{i-1}^-)$. Since $e_i = e_i^-$, we obtain $f(A_{i-1}^- + e_i^-) - f(A_{i-1}^-) = f(A_{i-1} + e_i) - f(A_{i-1})$. Putting everything together, for any $i \in [1, r)$ we have

$$w(e_i) = w^-(e_i) = w^-(e_i^-) = f(A_{i-1}^- + e_i^-) - f(A_{i-1}^-) = f(A_{i-1} + e_i) - f(A_{i-1}) = \Delta(e_i|A_{i-1}) \ .$$

**Weight invariant for DELETE($e$):** We next prove that the weight invariant holds upon deletion of an element $e$ at time $t$. According to Fact A.4, for any $i \in [1, r)$ we have ❶ $e_i = e_i^-$ ❷ $w(e_i) = w^-(e_i)$ ❸ $A_i = A_i^-$. In addition, for $i = 0$, we have $A_i = A_i^-$. From the definition of $r$ we know that $e_i^- \in R_i^- \setminus \{e\}$. Using Fact A.5, for any $i \in [0, s]$, it holds that $B_i = B_i^- \setminus \{v\}$ and for any $i \in [s + 1, r]$, we have $B_i = B_i^-$. Therefore, $B_i = B_i^- \setminus \{v\}$ which means that $e_i = e_i^- \in B_i^- \setminus \{e\} = B_i$.

By the assumption, we have $w^-(e_i^-) = \Delta(e_i^-|A_{i-1}^-) = f(A_{i-1}^- + e_i^-) - f(A_{i-1}^-)$. Thus, for any $i \in [1, r)$ we have $w(e_i) = w^-(e_i) = w^-(e_i^-) = f(A_{i-1}^- + e_i^-) - f(A_{i-1}^-) = f(A_{i-1} + e_i) - f(A_{i-1}) = \Delta(e_i|A_{i-1})$ which completes the proof that the weight invariant holds upon deletion of an element $e$ at time $t$.

## A.5  Maintenance of terminator invariant

Next, we prove that the terminator invariant holds. That is, upon the insertion or deletion of an arbitrary element $e$ at time $t$, we have $B_{i^*+1} = \emptyset$ assuming that $B_{i^{*-}+1}^- = \emptyset$ holds.

By the definition of $r$, if $r \leq i^{*-}$ we know that $r$ is the level where $e_r = e$ upon insertion or deletion of an element $e$ at time $t$, we know that Subroutines EXTENSION and RECURSION have been invoked. In addition, we have $s = r$. If this happens, RECURSION subroutine terminates when $|B_i| > 0$. As the result, we have $i^* = i - 1$ which means that $B_{i^*+1} = \emptyset$.

However, if $r = i^{*-} + 1$, we know that Subroutines EXTENSION and RECURSION have never been invoked for INSERT($e$) and DELETE($e$). Thus, the last level $i^*$ after inserting $e$ is the same last level $i^{*-}$ before inserting $e$. That is, $i^* = i^{*-}$. Note that, we must have $s \neq i^{*-} + 1$, since otherwise, we have $|B_{i^{*-}+1}^-| = 0$ and $|B_{i^*+1}| = 1$. Therefore, we have invoked Subroutines EXTENSION and RECURSION for this level what means $i^* > i^{*-}$ which is not the case. Thus, $s < r = i^{*-} + 1$ which means $B_{i^*+1} = B_{i^{*-}+1}^- = \emptyset$.

## A.6 Maintenance of uniform invariant

We now prove that the uniform invariant holds. That is, for any $i \leq i^*$ we have $\mathbb{P}[\mathbf{e}_i = e | \mathbf{i}^* \geq i \text{ and } \mathbf{C}_i = C_i] = \frac{1}{|B_i|} \cdot \mathbb{1}[e \in B_i]$. Consider the time when the element $e_i$ is sampled. At this time, all of the random values in $\mathbf{C}_i$ are realized. By construction, $e_i$ is sampled uniformly at random from $B_i$. Therefore, at this time the claim holds. Additionally, none of the values in $\mathbf{C}_i$ are changed afterwards in the execution. Therefore, the claim holds at the end of the execution as well.

In this section, we prove the uniform invariant, i.e., we show that for any $i \leq i^*$ we have $\mathbb{P}[\mathbf{e}_i = e | \mathbf{i}^* \geq i \text{ and } \mathbf{C}_i = C_i] = \frac{1}{|B_i|} \cdot \mathbb{1}[e \in B_i]$. We first that the invariant holds after calling INIT because we sample $e_i$ to be uniformly at random from $B_i$ and none of the value of $\mathbf{C}_i$ are changed after this sampling. The main thing to prove is that the invariant holds after each insertion and deletion operation.

Assume that the invariant holds before some update which is either the insertion or deletion of an element $v$. we will prove that it holds after the update as well. We use the superscript $^-$ to denote values before the update, e.g., $e_i^-$ denotes the value of $e_i$ before the update. When no superscript is used, we are referring to values after the update. By assumption, the invariant holds before the update, i.e., for any $i$ and $e$, and any $C_i$ such that $\mathbb{P}\left[\mathbf{i}^{*-} \geq i, \mathbf{C}_i^- = C_i\right] > 0$ we have we have

$$\mathbb{P}\left[\mathbf{e}_i^- = e | \mathbf{i}^{*-} \geq i, \mathbf{C}_i^- = C_i^-\right] = \frac{1}{|B_i^-|} \cdot \mathbb{1}\left[e \in B_i^-\right] \ . \tag{1}$$

We need to show that after the update, for any arbitrary $i$ and $e$, and any $C_i$ such that $\mathbb{P}[\mathbf{i}^* \geq i, \mathbf{C}_i = C_i] > 0$ we have

$$\mathbb{P}[\mathbf{e}_i = e | \mathbf{i}^* \geq i, \mathbf{C}_i = C_i] = \frac{1}{|B_i|} \cdot \mathbb{1}[e \in B_i] \ .$$

Note that $\mathbb{P}[\mathbf{e}_i = e | \mathbf{i}^* \geq i, \mathbf{C}_i = C_i]$, is only defined when $\mathbb{P}[\mathbf{i}^* \geq i, \mathbf{C}_i = C_i] > 0$, which means that given the input and considering the behavior of our algorithm including its random choices, it is possible to reach a state where $\mathbf{i}^* \geq i$ and $\mathbf{C}_i = C_i$.

### A.6.1 Insertions

If $v \notin B_0$, then it means that the data structure has ignored $v$, which in turn implies that $\mathbf{C}_i = \mathbf{C}_i^-$ and $\mathbf{e}_i = \mathbf{e}_i^-$, and $i^* = i^{*,-}$ which means the claim follows from the induction assumption. Formally, the following claim holds.

*Claim* A.6. If $v \notin B_0$, then events $\{\mathbf{i}^* \geq i, \mathbf{C}_i = C_i\}$ and $\left\{\mathbf{i}^{*,-} \geq i, \mathbf{C}_{i-1}^- = C_{i-1}, v \notin \mathbf{B}_0\right\}$ are equivalent. Furthermore, the event implies that $\mathbf{e}_i = \mathbf{e}_i^-$.

*Proof.* The first direction of equivalency is clear; if we have $\{\mathbf{i}^* \geq i, \mathbf{C}_i = C_i\}$ then the algorithm has ignored $v$ because otherwise we would have $v \in \mathbf{B_0} = B_0$. Therefore, we have $\mathbf{i}^{*,-} = \mathbf{i}^*$ and $\mathbf{C}_i^- = \mathbf{C}_i$. For the other direction, since $v \notin \mathbf{B_0}$ we know that the algorithm has ignored $v$ which means that $\mathbf{i}^{*,-} = \mathbf{i}^*$ and $\mathbf{C}_i^- = \mathbf{C}_i$ as before.

Since the event implies that the algorithm has ignored $v$, we have $\mathbf{e}_i^- = \mathbf{e}_i$ for all $i$. $\square$

Given the above claim, for all $C_i$ such that $v \notin B_0$,

$$\mathbb{P}[\mathbf{e}_i = e \mid \mathbf{i}^* \geq i, \mathbf{C}_i = C_i]$$
$$= \mathbb{P}\left[\mathbf{e}_i^- = e \mid \mathbf{i}^{*,-} \geq i, \mathbf{C}_{i-1}^- = C_{i-1}, v \notin \mathbf{B}_0\right]$$
$$= \mathbb{P}\left[\mathbf{e}_i^- = e \mid \mathbf{i}^{*,-} \geq i, \mathbf{C}_{i-1}^- = C_{i-1}\right]$$
$$= \frac{1}{|B_i|}\mathbb{1}[e \in B_i] \ ,$$

Here, for the second equality we have used the fact that the realization of $\mathbf{e}_i^-$ is independent of whether $v \in B_0$. This is because the algorithm decides whether or not $v \in B_0$ by flipping a coin with probability $1 - q$ afte $\mathbf{e}_i^-$ is realized.

We therefore focus on $C_i$ for which $v \in B_0$.

For any $i$, let $\mathbf{P}_i$ denote the random variable that is 1 if we invoke RECURSION$(A_j, B_j)$ for some $j \leq i$, and equals 0 otherwise. We first show that

$$\mathbb{P}\left[\mathbf{e}_i = e \mid \mathbf{i}^* \geq i, \mathbf{C}_i = C_i, \mathbf{P}_{i-1} = 1\right] = \frac{1}{|B_i|} \cdot \mathbb{1}\left[e \in B_i\right] \quad . \tag{2}$$

This is because $\mathbf{e}_i$ is obtained by sampling uniformly at random from $B_i$ and at the time of the sampling, the values in $\mathbf{C}_i$ are realized and they do not change afterwards.

We now consider the case where $\mathbf{P}_{i-1} = 0$. We claim that

$$\mathbb{P}\left[\mathbf{e}_i = e \mid \mathbf{i}^* \geq i, \mathbf{C}_i = C_i, \mathbf{P}_{i-1} = 0\right] = \frac{1}{|B_i|} \cdot \mathbb{1}\left[e \in B_i\right] \quad . \tag{3}$$

If $v \notin B_i$, then we have $\mathbf{B}_i = \mathbf{B}_i^-$ and $\mathbf{e}_i = \mathbf{e}_i^-$. Define $C_i^-$ as $C_i^- = (B_0 \backslash \{v\}, \ldots, B_i \backslash \{v\}, e_1, \ldots, e_{i-1})$.

Define the event $\sigma$ as

$$\sigma = \{i^{*,-} \geq i, \mathbf{C}_i^- = C_i^-, v \in \mathbf{B}_0, \mathbf{P}_{i-1} = 0\}$$

*Claim* A.7. For any $C_i$ such that $\mathbb{P}\left[i^* \geq i, \mathbf{C}_i = C_i, \mathbf{P}_{i-1} = 0\right] > 0$, the event $\sigma$ is equivalent to $\{i^* \geq i, \mathbf{C}_i = C_i, \mathbf{P}_{i-1} = 0\}$

*Proof.* Assume that the said event holds. We first note that $i^{*,-} \geq i$; if this is not the case then $\mathbf{P}_{i-1} = 0$ implies that $\mathbf{i}^* < i$ as well which is not possible. Additionally, we must have $\mathbf{C}_i = C_i^-$ because $\mathbf{P}_{i-1} = 0$ ensures that $\mathbf{e}_j = \mathbf{e}_j^-$ for all $j \leq i$ and the only difference between $\mathbf{B}_j$ and $\mathbf{B}_j^-$ is that the former may possibly contain $v$. Finally, we have $v \in \mathbf{B}_0$ given the assumption on $C_i$ made in the beginning of the proof. We note that since we assumed that $\mathbb{P}\left[i^* \geq i, \mathbf{C}_i = C_i, \mathbf{P}_{i-1} = 0\right] > 0$, this further means that $\mathbb{P}\left[\sigma, \mathbf{C}_i = C_i, \right] > 0$.

For the converse, assume that $\sigma$ holds. Since $\mathbf{P}_{i-1} = 0$, we have $\mathbf{i}^* = \mathbf{i}^{*,-}$. We therefore need to show that $\mathbf{C}_i = C_i$. The values in $\mathbf{e}_1, \ldots, \mathbf{e}_{i-1}$ are not changed so $\mathbf{e}_j = \mathbf{e}_j^- = e_j$ by definition of $C_i^-$. Additionally, the values of $\mathbf{B}_0, \ldots, \mathbf{B}_i$ are now deterministic because we have fixed the realization of the coin flips occurring with probability $\frac{1}{|B_j|}$ by assuming $\mathbf{P}_{i-1} = 0$, and we have assumed $v \in \mathbf{B}_0$. Therefore, there is only one possible value for $\mathbf{C}_i$. Therefore, either $\mathbb{P}\left[\mathbf{C}_i = C_i \mid \sigma\right] = 0$ or $\mathbb{P}\left[\mathbf{C}_i = C_i \mid \sigma\right] = 1$. We have already proved that $\mathbb{P}\left[\mathbf{C}_i = C_i, \sigma\right] > 0$, which means we must have $\mathbb{P}\left[\mathbf{C}_i = C_i \mid \sigma\right] = 1$, finishing the proof. $\qquad \square$

*Claim* A.8. $\mathbb{P}\left[\mathbf{P}_i = 0 \mid \sigma\right] = \frac{\mathbb{1}\left[v \in B_i\right]}{|B_i|}$

*Proof.* If $\sigma$ holds then $\mathbf{P}_{i-1} = 0$ which means that $\mathbf{P}_i = 0$ if and only if the algorithm flips the coin in Line 2 to invoke recursion. If $v \notin B_i$, then this cannot happen because Line 2 would never be executed. If $v \in B_i$, then we note that this happens with probability $\frac{1}{|B_i|}$ because of Line 2. Note that at the time this line is executed, the values of $\mathbf{i}^{*,-}, \mathbf{C}_i^-, \mathbf{B}_0, \mathbf{P}_{i-1}$. $\qquad \square$

Combining the above claims we obtain

$$\begin{aligned}
&\mathbb{P}\left[\mathbf{e}_i = e \mid i^* \geq i, \mathbf{C}_i = C_i, \mathbf{P}_{i-1} = 0\right] \\
&= \mathbb{P}\left[\mathbf{e}_i = e \mid \sigma\right] && \textit{(Claim A.9)} \\
&= \frac{\mathbb{1}\left[v \in B_i\right]}{|B_i|} \mathbb{P}\left[\mathbf{e}_i = e \mid \sigma, \mathbf{P}_i = 1\right] + \left(1 - \frac{\mathbb{1}\left[v \in B_i\right]}{|B_i|}\right) \mathbb{P}\left[\mathbf{e}_i = e \mid \sigma, \mathbf{P}_i = 0\right] && \textit{(Claim A.9)} \\
&= \frac{\mathbb{1}\left[v \in B_i\right]}{|B_i|} \mathbb{1}\left[e = v\right] + \left(1 - \frac{\mathbb{1}\left[v \in B_i\right]}{|B_i|}\right) \mathbb{P}\left[\mathbf{e}_i = e \mid \sigma, \mathbf{P}_i = 0\right], && (4)
\end{aligned}$$

where for the final equality we have used the fact that if RECURSION$(A_i, B_i)$ is invoked (i.e., if $\mathbf{P}_i = 1$), then we must have $\mathbf{e}_i = v$. We next calculate $\mathbb{P}\left[\mathbf{e}_i = e \mid \sigma, \mathbf{P}_i = 0\right]$. We begin by observing that if $\mathbf{P}_i = 0$, then we must have $\mathbf{e}_i = \mathbf{e}_{i-1}$ which means

$$\mathbb{P}\left[\mathbf{e}_i = e \mid \sigma, \mathbf{P}_i = 0\right] = \mathbb{P}\left[\mathbf{e}_i^- = e \mid \sigma, \mathbf{P}_i = 0\right], \tag{5}$$

Define the event $\sigma'$ as $\sigma' = \left\{i^{*,-} \geq i, \mathbf{C}_i = C_i^-, v \in \mathbf{B}_0\right\}$, i.e., $\sigma'$ satisfies $\sigma = \{\sigma', \mathbf{P}_{i-1} = 0\}$.

*Claim* A.9. Conditioned on the event $\left\{i^{*,-} \geq i, \mathbf{C}_i = C_i^-\right\}$, the events $\{v \in \mathbf{B}_0, \mathbf{P}_i = 0\}$ and $\mathbf{e}_i^- = e$ are independent.

*Proof.* In order to have $v \in \mathbf{B}_0$ and $\mathbf{P}_i = 0$, the insertion algorithm should not ignore $v$ and for all $j$ such that we reach line 2, we need to flip the coin such that RECURSION is not invoked. The realization of $\mathbf{e}_i^-$ has no effect on this. For $\mathbf{B}_0$, the coint is always flipped with the same probability and for $\mathbf{P}_i = 0$, if the coins are flipped, they are always flipped with the same probability $\frac{1}{|B_j|}$, and whether or not the coin is flipped (i.e., Line 2 is executed) depends only the value of $\mathbf{B}_j$ and $\mathbf{A}_{j-1}$, neither of which are affected by the realization of $\mathbf{e}_i^-$. $\qquad\square$

Given the above claim,

$$\mathbb{P}\left[\mathbf{e}_i^- = e \mid \sigma, \mathbf{P}_i = 0\right]$$
$$= \mathbb{P}\left[\mathbf{e}_i = e \mid i^{*,-} \geq i, \mathbf{C}_i = C_i^-, v \in \mathbf{B}_0, \mathbf{P}_i = 0\right] \qquad \text{(Definition of $\sigma$)}$$
$$= \mathbb{P}\left[\mathbf{e}_i^- = e \mid i^{*,-} \geq i, \mathbf{C}_i = C_i^-\right] \qquad \text{(Claim A.9)}$$
$$= \frac{\mathbb{1}\left[e \in B_i^-\right]}{\left|B_i^-\right|} \qquad \text{(Induction assumption (1))}$$
$$= \frac{\mathbb{1}\left[e \in B_i \setminus \{v\}\right]}{\left|B_i \setminus \{v\}\right|} \qquad \text{(Definition of $B_i$)}$$

Combined with Equation (4) and Equation (5), this gives us

$$\mathbb{P}\left[\mathbf{e}_i = e \mid i^* \geq i, \mathbf{C}_i = C_i, \mathbf{P}_i = 0\right]$$
$$= \frac{\mathbb{1}\left[v \in B_i, e = v\right]}{|B_i|} + \left(1 - \frac{\mathbb{1}\left[v \in B_i\right]}{|B_i|}\right) \frac{\mathbb{1}\left[e \in B_i \setminus \{v\}\right]}{|B_i \setminus \{v\}|}.$$

We therefore need to show that the above expression is the same as $\frac{\mathbb{1}[e \in B_i]}{|B_i|}$. This is easy to check however using a case by case analysis. If $v \notin B_i$, then the first term vanishes and the second term becomes $\frac{\mathbb{1}[e \in B_i]}{|B_i|}$ as required. If $v \in B_i$ then the expression can be rewritten as

$$\mathbb{P}\left[\mathbf{e}_i = e \mid i^* \geq i, \mathbf{C}_i = C_i, \mathbf{X}_i = 0\right] = \frac{\mathbb{1}\left[e = v\right]}{|B_i|} + \frac{|B_i| - 1}{|B_i|} \frac{\mathbb{1}\left[e \in B_i \setminus \{v\}\right]}{|B_i| - 1}$$
$$= \frac{\mathbb{1}\left[e = v\right]}{|B_i|} + \frac{\mathbb{1}\left[e \in B_i \setminus \{v\}\right]}{|B_i|}.$$

Depending on whether $e \neq v$ or $e = v$, either the first term or (respectively) the second term disappears and the other term equals $\frac{1}{|B_i|}$. This proves Equation (5) which combined with Equation (4) finishes the proof.

### A.6.2 Deletion

For any $i$, let $\mathbf{P}_i$ denote the random variable that is 1 if we invoke RECURSION($A_j, B_j$) for some $j \leq i$, and equals 0 otherwise. We first claim that

$$\mathbb{P}\left[\mathbf{e}_i = e \mid \mathbf{i}^* \geq i, \mathbf{C}_i = C_i, \mathbf{P}_i = 1\right] = \frac{1}{|B_i|} \cdot \mathbb{1}\left[e \in B_i\right]. \qquad (6)$$

As in the case of insertion, this is because $\mathbf{e}_i$ is obtained by sampling uniformly at random from $B_i$ and at the time of the sampling, the values in $\mathbf{C}_i$ are realized and they do not change afterwards.

We next show that

$$\mathbb{P}\left[\mathbf{e}_i = e \mid \mathbf{i}^* \geq i, \mathbf{C}_i = C_i, \mathbf{P}_i = 0\right] = \frac{1}{|B_i|} \cdot \mathbb{1}\left[e \in B_i\right]. \qquad (7)$$

We start by evaluating the left hand side. We first claim that the event $\{\mathbf{i}^* \geq i, \mathbf{C}_i = C_i, \mathbf{P}_i = 0\}$ implies the event $\mathbf{i}^{*,-} \geq i$. This is because if $\mathbf{i}^{*,-} < i$, then $\mathbf{P}_i = 0$ means that RECURSION is not invoked by deletion, which in turn means that $\mathbf{i}^* = \mathbf{i}^{*,-}$, contradicting the assumption $\mathbf{i}^* \geq i$. Since $\mathbf{i}^{*,-} \geq i$, the value of $\mathbf{C}_i^-$ is defined. We condition on the value of $\mathbf{C}_i^-$. Formally,

$$\mathbb{P}\left[\mathbf{e}_i = e \mid \mathbf{i}^* \geq i, \mathbf{C}_i = C_i, \mathbf{P}_i = 0\right] = \mathbb{E}_{C_i^-}\left[\mathbb{P}\left[\mathbf{e}_i = e \mid \mathbf{i}^* \geq i, \mathbf{C}_i = C_i, \mathbf{P}_i = 0, \mathbf{C}_i^- = C_i^-\right]\right], \qquad (8)$$

where in the above expectation, the value of $C_i^-$ is sampled from the distribution of $\mathbf{C}_i^-$ conditioned on $\{\mathbf{i}^* \geq i, \mathbf{C}_i = C_i, \mathbf{P}_i = 0\}$. We note that all the values considered in the expectation satisfy $\mathbb{P}\left[\mathbf{i}^* \geq i, \mathbf{C}_i = C_i, \mathbf{P}_i = 0, \mathbf{C}_i^- = C_i^-\right] > 0$.

Consider some $C_i^-$ such that $\mathbb{P}\left[\mathbf{i}^* \geq i, \mathbf{C}_i = C_i, \mathbf{P}_i = 0, \mathbf{C}_i^- = C_i^-\right]$. We claim that rewrite the event $\left\{\mathbf{i}^* \geq i, \mathbf{C}_i = C_i, \mathbf{P}_i = 0, \mathbf{C}_i^- = C_i^-\right\}$ is equivalent to $\left\{\mathbf{i}^{*,-} \geq i, \mathbf{C}_i^- = C_i^-, \mathbf{P}_i = 0\right\}$. It is clear that the first event implies the second event. As for the other direction, we note that the realization of $\mathbf{C}_i^-$ fully determines the realization of $\mathbf{C}_i$; specifically, since $\mathbf{P}_i = 0$ we have $\mathbf{B}_j = \mathbf{B}_j^- \setminus \{v\}$ for all $j \leq i$ and $\mathbf{e}_j = \mathbf{e}_j^-$ for all $j < i$. Therefore, $\mathbb{P}\left[\mathbf{i}^* \geq i, \mathbf{C}_i = C_i, \mathbf{P}_i = 0 \mid \mathbf{i}^{*,-} \geq i, \mathbf{C}_i^- = C_i^-\right]$ is either 0 or 1. We know that it cannot be 0 however because by assumption, $\mathbb{P}\left[\mathbf{i}^* \geq i, \mathbf{C}_i = C_i, \mathbf{P}_i = 0, \mathbf{C}_i^- = C_i^-\right] > 0$. Therefore, it must be 1 which means $\mathbf{C}_i = C_i$. Finally, since $\mathbf{P}_i = 0$ we have $\mathbf{i}^* = \mathbf{i}^{*,-} \geq i$. We note that $\mathbf{P}_i = 0$ itself implies that $\mathbf{e}_i = \mathbf{e}_i^-$. Therefore, we can rewrite the probability inside the expectation in Equation (8) as

$$\mathbb{P}\left[\mathbf{e}_i = e \mid \mathbf{i}^* \geq i, \mathbf{C}_i = C_i, \mathbf{P}_i = 0, \mathbf{C}_i^- = C_i^-\right]$$
$$= \mathbb{P}\left[\mathbf{e}_i^- = e \mid \mathbf{i}^{*,-} \geq i, \mathbf{C}_i^- = C_i^-, \mathbf{P}_i = 0\right]$$
$$= \frac{\mathbb{P}\left[\mathbf{P}_i = 0 \mid \mathbf{e}_i^- = e, \mathbf{i}^{*,-} \geq i, \mathbf{C}_i^- = C_i^-\right] \mathbb{P}\left[\mathbf{e}_i^- = e \mid \mathbf{i}^{*,-} \geq i, \mathbf{C}_i^- = C_i^-\right]}{\mathbb{P}\left[\mathbf{P}_i = 0 \mid \mathbf{i}^{*,-} \geq i, \mathbf{C}_i^- = C_i^-\right]} \qquad \textit{(Bayes' Rule)}$$

We first claim that $v \neq e_j^-$ for all $j < i$. This is because if $v = e_j^-$ for some $j < i$, then $\mathbb{P}\left[\mathbf{P}_i = 0 \mid \mathbf{i}^{*,-} \geq i, \mathbf{C}_i^- = C_i^-\right] = 0$ because the deletion of $v$ (which is $e_j^-$) would invoke RECURSION. By assumption however, we only considered values of $C_i^-$ for which $\mathbb{P}\left[\mathbf{C}_i^- = C_i^-, \mathbf{P}_i = 0\right] > 0$ in the expectation in Equation (8).

Given this, we have $\mathbf{P}_i = 0$ if and only if $v \neq \mathbf{e}_i^-$. It follows that

$$\mathbb{P}\left[\mathbf{P}_i = 0 \mid \mathbf{i}^{*,-} \geq i, \mathbf{C}_i^- = C_i^-\right] = \mathbb{P}\left[\mathbf{e}_i^- \neq v \mid \mathbf{i}^{*,-} \geq i, \mathbf{C}_i^- = C_i^-\right] = 1 - \frac{\mathbb{1}\left[v \in B_i^-\right]}{\left|B_i^-\right|}.$$

and

$$\mathbb{P}\left[\mathbf{P}_i = 0 \mid \mathbf{e}_i^- = e, \mathbf{i}^{*,-} \geq i, \mathbf{C}_i^- = C_i^-\right] = \mathbb{P}\left[\mathbf{e}_i^- \neq v \mid \mathbf{e}_i^- = e, \mathbf{i}^{*,-} \geq i, \mathbf{C}_i^- = C_i^-\right] = \mathbb{1}\left[e \neq v\right].$$

By induction hypothesis (Equation (1)),

$$\mathbb{P}\left[\mathbf{e}_i^- = e \mid \mathbf{i}^{*,-} \geq i, \mathbf{C}_i^- = C_i^-\right] = \frac{\mathbb{1}\left[e \in B_i^-\right]}{\left|B_i^-\right|}$$

Putting it all together we obtain

$$\mathbb{P}\left[\mathbf{e}_i = e \mid \mathbf{i}^* \geq i, \mathbf{C}_i = C_i, \mathbf{P}_i = 0, \mathbf{C}_i^- = C_i^-\right] = \frac{\mathbb{1}\left[e \in B_i^-\right] \mathbb{1}\left[e \neq v\right]}{\left|B_i^-\right| \left(1 - \frac{\mathbb{1}\left[v \in B_i^-\right]}{\left|B_i^-\right|}\right)}.$$

If $v \notin B_i^-$, then we have $B_i = B_i^-$ and the above expression turns into

$$\frac{\mathbb{1}\left[e \in B_i\right] \mathbb{1}\left[e \neq v\right]}{\left|B_i\right|} = \frac{\mathbb{1}\left[e \in B_i\right]}{\left|B_i\right|},$$

where for the equality we have used the fact that if $e \in B_i$ we must have $e \neq v$ since $v \notin B_i$. If $v \in B_i$, then the above expression becomes

$$\frac{\mathbb{1}\left[e \in B_i \cup \{v\}\right] \mathbb{1}\left[e \neq v\right]}{\left(\left|B_i\right| + 1\right)\left(1 - \frac{1}{\left|B_i\right|+1}\right)} = \frac{\mathbb{1}\left[e \in B_i\right]}{\left|B_i\right|},$$

where for the equality we have used the fact that $e \in B_i \{v\}$ and $e \neq v$ is equivalent to $e \in B_i$ to rewrite the numerator and simplified $\left(\left|B_i\right| + 1\right)\left(1 - \frac{1}{\left|B_i\right|+1}\right)$ to

$$\left(\left|B_i\right| + 1\right) \frac{\left|B_i\right|}{\left|B_i\right| + 1} = \left|B_i\right|,$$

to rewrite the denominator. We have therefore proved Equation (7) which finishes the proof together with Equation (6).

## A.7   Query complexity

To analyze the query complexity of this algorithm, we first prove that checking if an element $e$ can be added to level $i$ requires $O(p \cdot \log(k))$ oracle queries due to Lemma A.10. Consequently, the FILTERING algorithm requires $O(|B_j| \cdot p \cdot \log(k))$ queries to operate on level $j$.

Therefore, running RECURSION on level $i$ requires $O(i^* \cdot |B_i| \cdot p \cdot \log(k))$ oracle calls to construct levels $i, i+1, \ldots, i^*$. Given the uniform invariant, in each update, we reconstruct at level $i$ with probability $1/B_i$. Thus, the expected number of oracle calls for reconstructing at level $i$ is $O(i^* p \log(k))$.

Summing this up for all levels, the expected number of oracle calls is $O((i^*)^2 p \log(k))$, which can be written as $O(\varepsilon^{-2} p k^4 \log(k))$ because of Lemma A.11.

Finally, as mentioned in Section B.2, each element is applied in $O(\log(\frac{k}{\varepsilon}))$ parallel runs, resulting in a total query complexity of $O(\varepsilon^{-3} p k^4 \log^2(k))$.

**Lemma A.10** (Lemma 3.2 in Banihashem et al. [2024]). *Let $I \in \mathcal{I}_j$ be an independent set of matroid $\mathcal{M}_i(\mathcal{V}_i, \mathcal{I}_i)$ and $e$ be an element such that $I \cup \{e\} \notin \mathcal{I}_j$. Define $X_j := \{e' : I - e' + e \in \mathcal{I}_j\}$. Let $w : I \cup \{e\} \to \mathbb{R}^{\geq 0}$ be an arbitrary weight function and define $x_j := \arg\min_{e' \in X_j} w(e')$. The element $x_j$ can be found using at most $O(\log(k))$ oracle queries.*

**Lemma A.11.** *The number of levels $i^*$ is at most $O\left(\frac{k^2}{\varepsilon}\right)$.*

*Proof.* For each element $e \in \mathcal{V}$, we have $f(e) \leq \text{MAX}$. Therefore, because of submodularity, we have $w(e) = f(e|A_i) \leq f(e) \leq \text{MAX}$. Thus, we can conclude that $w(A_{i^*}) = \sum_{e \in A_{i^*}} w(e) \leq k \cdot \text{MAX}$.

Moreover, if at level $i$, we add element $e_i$, we have two cases:

- If $e_i$ does not replace any other elements, given the survivor invariant, $w(e_i) \geq \frac{\varepsilon \cdot \text{MAX}}{k}$, which means $w(A_i) - w(A_{i-1}) \geq \frac{\varepsilon \cdot \text{MAX}}{k}$.

- If the element removes a set of elements $U_i(e_i)$, its weight should be larger than $(1 + \sqrt{1 + \frac{1}{p}})$ times the total weight of the removed elements in $U_i(e_i)$. This means removing elements of $U_i(e_i)$ and adding $e_i$ adds at least $\sqrt{1 + \frac{1}{p}} \cdot w(U_i(e_i)) \geq \sqrt{1 + \frac{1}{p}} \cdot \frac{\varepsilon \cdot \text{MAX}}{k} \geq \frac{\varepsilon \cdot \text{MAX}}{k}$.

Since after each level, the value of $w(A_i)$ increases by at least $\frac{\varepsilon \cdot \text{MAX}}{k}$ and it cannot exceed $k \cdot \text{MAX}$, the number of levels is at most

$$\frac{k \cdot \text{MAX}}{\frac{\varepsilon \cdot \text{MAX}}{k}} = \frac{k^2}{\varepsilon}.$$

$\square$

Next, we analyze the query complexity of calling RECURSION on level $i$.

**Lemma A.12.** *The total cost of calling RECURSION$(i)$ is at most $O(\varepsilon^{-1} p k^2 \cdot |B_i| \cdot \log(k))$.*

*Proof.* Based on Lemma A.10, inside the for loop of DEPENDENCYDETECTION requires $O(\log(k))$ query calls to find a replacing element. Since each element is inside at most $p$ matroids, we need to run inside that for loop $p$ times, which makes the query complexity of DEPENDENCYDETECTION $O(p \cdot \log(k))$.

Since FINDSWAPS calls DEPENDENCYDETECTION, and FILTERING calls FINDSWAPS on every element of $|B_i|$, one step of RECURSION needs $O(|B_i| \cdot p \cdot \log(k))$ oracle calls.

As we proved in Lemma A.11, the depth of recursion in RECURSION is at most $O\left(\frac{k^2}{\varepsilon}\right)$. Therefore, the query complexity of RECURSION$(i)$ is $O\left(\varepsilon^{-1} k^2 \cdot |B_i| \cdot p \cdot \log(k)\right)$. $\square$

**Lemma A.13.** *For a specified value of MAX, each update operation in Algorithm UPDATEOPERATIONS has query complexity at most $O(\varepsilon^{-2} p k^4 \log(k))$.*

*Proof.* Based on the uniform invariant, when we insert or delete an element, for each level $i \leq i^*$, we call RECURSION$(i)$ with probability $\frac{1}{|B_i|} \cdot \mathbb{1}[e \in B_i]$, which is at most $\frac{1}{|B_i|}$. Using Lemma A.12, the

query complexity for calling RECURSION($i$) is $O(\varepsilon^{-1}pk^2 \cdot |B_i| \cdot \log(k))$. Therefore, the expected number of queries caused by level $i$ is bounded by

$$\frac{1}{|B_i|} \cdot O\left(\varepsilon^{-1}pk^2 \cdot |B_i| \cdot \log(k)\right) = O\left(\varepsilon^{-1}pk^2 \log(k)\right).$$

As Lemma A.11 bounds the number of levels by $i^* = O\left(\frac{k^2}{\varepsilon}\right)$, we calculate the expected number of query calls for each update by summing the expected number of query calls at each level:

$$\sum_{i=1}^{i^*} O\left(\varepsilon^{-1}pk^2 \log(k)\right) \le O\left(\varepsilon^{-2}pk^4 \log(k)\right).$$

□

To create an algorithm that operates independently of the value of $MAX$, we employ a strategy of guessing $MAX$ up to a factor of 2 using parallel runs. Each element is inserted into only $\log(k/\varepsilon)$ instances of the algorithm. Consequently, we achieve the total query complexity as claimed in Theorem 5.4.

## A.8 Approximation guarantee

In this section, we aim to establish the approximation guarantee of our algorithm. For this purpose, we make the assumption that elements are not randomly discarded at the beginning. Instead, we discard them with probability $1 - q$ when attempting to add them to our solution. While this adjustment does not affect the solution itself, it simplifies the proof of the approximation factor. We denote the set of elements discarded in this approach as $R$, and we use $M$ to represent the set of elements filtered in FINDSWAPS because their marginal gain was less than $\frac{\varepsilon MAX}{k}$.

Before starting the proofs, we introduce some useful variables and notations. First, we define $A' := \cup_{i=1}^{i^*} A_i$ as the union of solutions at all levels. Additionally, we introduce a crucial notation for our proofs:

**Definition A.14.** Let $e \in \mathcal{V}$ be an element and $S \subseteq \mathcal{V}$ be a set of elements. Let $i \ge 0$ be the largest number such that $e \in B_i$. We define $f(e : S) := \Delta(e|S \setminus B_i)$, which calculates the marginal gain of adding $e$ to a subset of $S$ containing only elements that have been selected or filtered in levels preceding the level of selecting or filtering $e$. Note that $w(e) = \Delta(e|A_{i-1}) = \Delta(e|A_{i-1} \setminus B_i) = f(e : A_{i-1})$, where $w$ is the weight function defined in Algorithm 1. For sets $T, S \subseteq \mathcal{V}$, we define $f(T : S) := \sum_{e \in T} f(e : S)$.

Additionally, for every element $e \in \mathcal{V}$, we define $d(e)$ as the first level $i$ such that $e \notin A_i \cup B_i$.

Now we represent several lemmas from Feldman et al. [2018] and utilize them in our proofs. In the following, we use the constant $c$, defined as $c := \sqrt{1 + \frac{1}{p}}$.

**Lemma A.15** (Lemma 7 in Feldman et al. [2018]). $f(A' \setminus A_{i^*} : A_{i^*}) \le \dfrac{f(A_{i^*})}{c}$.

**Lemma A.16** (Corollary 8 in Feldman et al. [2018]). $f(A') \le \dfrac{c+1}{c} \cdot f(A_{i^*})$.

**Lemma A.17** (Proposition 10 in Feldman et al. [2018]). *For every independent set $S \subseteq \mathcal{V} \setminus (M \cup R)$, there exists a mapping $\phi_S$ from elements of $S$ to multi-subsets of $A'$ such that:*

- *Each element $e \in A_{i^*}$ appears at most $p$ times in the multi-sets of $\{\phi_S(e) \mid e \in S\}$.*

- *Each element $e \in A' \setminus A_{i^*}$ appears at most $p - 1$ times in the multi-sets of $\{\phi_S(e) \mid e \in S\}$.*

- *Each element $e \in S \setminus A'$ satisfies $w(e) \le (1 + c) \cdot \sum_{e' \in \phi_S(e)} f(e' : A_{d(e')-1})$.*

- *Each element $e \in S \cap A'$ satisfies $w(e) \le f(e' : A_{d(e')-1})$ for every $e' \in \phi_S(e)$, and the multi-set $\phi_S(e)$ contains exactly $p$ elements (including repetitions).*

**Theorem A.18.** $\mathbb{E}\left[f(A' \cup OPT)\right] \le \frac{(1+c)^2 p}{c} \cdot \mathbb{E}\left[f(A_{i^*})\right] + \varepsilon f(OPT)$.

*Proof.* First, we break $A' \cup OPT$ into a union of four sets:

$$f(A' \cup OPT) \leq f(A') + \sum_{OPT \setminus (A' \cup R \cup M)} \Delta(e|A') + \sum_{OPT \cap R} \Delta(e|A') + \sum_{OPT \cap M} \Delta(e|A') \qquad \text{(submodularity)}$$

$$\leq f(A') + \sum_{OPT \setminus (A' \cup R \cup M)} w(e) + \sum_{OPT \cap R} w(e) + \sum_{OPT \cap M} w(e) \qquad (w(e) = \Delta(e|A_i) \geq \Delta(e|A'))$$

$$\leq \frac{1+c}{c} \cdot f(A_{i^*}) + \sum_{OPT \setminus (A' \cup R \cup M)} w(e) + \sum_{OPT \cap R} w(e) + \sum_{OPT \cap M} w(e). \qquad \text{(Lemma A.16)}$$

If we set $S = OPT \setminus (R \cup M)$, we can use Lemma A.17 to bound the second term of the last expression:

$$\sum_{OPT \setminus (A' \cup R \cup M)} w(e) \leq (1+c) \cdot \sum_{\substack{e \in OPT \setminus (A' \cup R \cup M) \\ e' \in \phi_{OPT \setminus (R \cup M)}(e)}} f(e' : S_{d(e')-1}).$$

Moreover,

$$\sum_{\substack{e \in OPT \setminus (A' \cup R \cup M) \\ e' \in \phi_{OPT \setminus (R \cup M)}(e)}} f(e' : S_{d(e')-1}) + p \cdot \sum_{e \in OPT \cap A'} w(e)$$

$$\leq \sum_{\substack{e \in OPT \setminus A' \\ e' \in \phi_{OPT \setminus (R \cup M)}(e)}} f(e' : S_{d(e')-1}) \qquad \text{(Lemma A.17)}$$

$$\leq p \cdot \sum_{e \in A_{i^*}} f(e : A_{i^*}) + (p-1) \cdot \sum_{e \in A' \setminus A_{i^*}} f(e : S_{d(e)-1}) \qquad \text{(Lemma A.17)}$$

$$\leq p \cdot f(A_{i^*}) + \frac{p-1}{c} \cdot f(A_{i^*}) \qquad \text{(Lemma A.15)}$$

$$= \frac{(1+c)p - 1}{c} \cdot f(A_{i^*}).$$

Combining all together, we get:

$$f(A' \cup OPT) \leq \frac{1+c}{c} \cdot f(A_{i^*}) + (1+c) \cdot \left[ \frac{(1+c) \cdot p - 1}{c} \cdot f(A_{i^*}) - p \sum_{e \in OPT \cap A'} w(e) \right]$$

$$+ \sum_{OPT \cap R} w(e) + \sum_{OPT \cap M} w(e)$$

$$= \frac{(1+c)^2 p}{c} \cdot f(A_{i^*}) - (1+c)p \sum_{e \in OPT \cap A'} w(e) + \sum_{e \in OPT \cap R} w(e) + \sum_{OPT \cap M} w(e).$$

We can bound the last term given that for each element $e \in M$, we have $w(e) \leq \frac{\varepsilon MAX}{k} \leq \frac{\varepsilon f(OPT)}{k}$ and $|OPT| \leq k$, we can conclude that

$$\sum_{e \in OPT \cap M} w(e) \leq \varepsilon f(OPT).$$

Now, we want to show that the expectations of the second term and the third term are equal. Consider an arbitrary element $e \in OPT$. Based on our assumption discussed at the beginning of this section about discarding elements randomly, if they were going to be added to the solution instead of being discarded at the beginning, each time an element is considered for addition to the solution, with probability $q$ it will be added to $A'$ and with probability $1 - q$ it will be discarded, which means it will be added to $R$. Therefore, we have

$$\frac{\mathbb{E}[\mathbb{1}[e \in A'] \cdot w(e)]}{q} = \frac{\mathbb{E}[\mathbb{1}[e \in R] \cdot w(e)]}{1-q},$$

where $\mathbb{1}[X]$ is set to one if $X$ holds and is set to zero otherwise. Rearranging the above equality, we have

$$\mathbb{E}\left[ \sum_{e \in OPT \cap R} w(e) \right] = \frac{1-q}{q} \cdot \mathbb{E}\left[ \sum_{e \in OPT \cap A'} w(e) \right] = (1+c)p \cdot \mathbb{E}\left[ \sum_{e \in OPT \cap A'} w(e) \right],$$

where the last equality comes from the fact that the equality $\frac{1}{q} - 1 = (1 + c)p$ holds for $q = (p + \sqrt{p^2 + p} + 1)^{-1}$ and $c = \sqrt{1 + \frac{1}{p}}$.

By combining the above, we get

$$f(A' \cup OPT) \leq \frac{(1 + c)^2 p}{c} \cdot f(A_{i^*}) + \varepsilon f(OPT).$$

$\square$

**Lemma A.19** (Lemma 2.2 in Buchbinder et al. [2014]). *Let $g : 2^{\mathcal{V}} \to \mathbb{R}^+$ be a submodular function. Let $S$ be a random subset of $\mathcal{V}$ such that each element appears in $S$ with probability at most $q$, not necessarily independently. We have $\mathbb{E}[g(S)] \geq (1 - q)g(\emptyset)$.*

**Proof of Theorem 5.5:** Let's define $g(S) = f(S \cup OPT)$. Since $g$ is a non-negative submodular function, and each element will be discarded with probability $1 - q$, meaning it can appear in $A'$ with probability at most $q$, we can use Lemma A.19 to conclude that

$$\mathbb{E}[f(A' \cup OPT)] = \mathbb{E}[g(A')] \geq (1 - q)g(\emptyset) = (1 - q)f(OPT) = \frac{1}{c}f(OPT),$$

where the last equality holds because the equation $1 - q = \frac{1}{c}$ for $q = (p + \sqrt{p^2 + p} + 1)^{-1}$ and $c = \sqrt{1 + \frac{1}{p}}$. Combining this with Theorem A.18, we get

$$\frac{(1 + c)^2 p}{c} \cdot \mathbb{E}[f(A_{i^*})] + \varepsilon f(OPT) \geq \frac{1}{c} \cdot f(OPT),$$

or rearranging it as

$$\mathbb{E}[f(A_{i^*})] \geq \frac{c}{(1 + c)^2 p} \left( \frac{1}{c} - \varepsilon \right) f(OPT) = \left( \frac{1}{2p + 2\sqrt{p(p + 1)} + 1} - \varepsilon' \right) f(OPT).$$

$\square$

**Proof of Proposition 1.3:** The proof of this result is analogous to the proof of the main result 1.1, with the key difference that its approximation guarantee follows directly from Theorem A.18, which relies solely on the equation $\frac{1}{q} - 1 = (1 + c)p$, unlike the proof of Theorem 5.5, which also uses the equation $1 - q = \frac{1}{c}$. This flexibility allows the approximation ratio to be optimized by adjusting the parameters and setting $c = 1$ and $q = (2p + 1)^{-1}$.

$\square$

# B  Removing known MAX assumption

In this section, we show how to remove the assumption that we know a parameter MAX satisfying $\max_{v \in \mathcal{V}} f(v) \leq \text{MAX} \leq 2 \max_{v \in \mathcal{V}} f(v)$. While the techniques here are standard in the literature for both dynamic and streaming Kazemi et al. [2019], Lattanzi et al. [2020], Banihashem et al. [2024] submodular optimization, we here provide the approach and the proof for completeness.

## B.1  Overview of the reduction

Let $\mathcal{A}$ denote a dynamic algorithm that operates without this knowledge, i.e., the algorithm presented in the paper. We will maintain parallel runs of the algorithm $\mathcal{A}$ in memory which try to guess the value MAX. Formally, for any integer $i$, possibly negative, let $\mathcal{A}_i$ denote a run of the algorithm $\mathcal{A}$ with the parameter MAX set to $2^i$. Note that we will not actually initialize these runs for all $i$ and will only do so for some values based on the inserted elements.

When an element $v$ is inserted, define $I_v$ as

$$I_v := \left\{ i : \frac{\varepsilon}{k} 2^i \leq f(v) \leq 2^i \right\}$$

For each $i \in I_v$, if the instance $\mathcal{A}_i$ is not initialized, then we initialize it. Next, for all $i \in I_v$, we insert $v$ in $\mathcal{A}_i$.
When an element is deleted, we delete it from all $\mathcal{A}_i$ for $i \in I_v$.
We always output the maximum answer among all parallel runs that are initialized. We note that this does not require any queries.

## B.2 Analysis of query complexity

Since $|I_v| \leq \log(k/\varepsilon)$, and initializing $\mathcal{A}_i$ with an empty ground set does not require any queries, the expected query complexity of the algorithm is $O(\log(k/\varepsilon))$ times the expected query complexity of $\mathcal{A}_i$.

## B.3 Analysis of approximation guarantee

We first consider an alternative version of the algorithm in which we set $I_v$ to be

$$I_v = \left\{ i : f(v) \leq 2^i \right\}$$

We claim that the output of this version is the same as the version described in Section B.1. Note that this new algorithm is used for the purpose of analysis only and is not actually used for implementation.

This is because if $f(v) < \frac{\varepsilon}{k} 2^i$, then the algorithm simply discards it given the check in FindSwaps. For the new version of the algorithm, set $i'$ such that $2^{i'-1} \leq f(v) \leq 2^i$. Given the analysis in Section A, the output of $\mathcal{A}_{i'}$ provides the desired approximation guarantee. Since we are always taking the output of the $\mathcal{A}_i$ with the best output, our output provides the desired approximation guarantee as well.

