# OpenReview forum: "Non-monotone Submodular Optimization: $p$-Matchoid Constraints and Fully Dynamic Setting"
_NeurIPS.cc/2025/Conference — NeurIPS 2025 poster_

### Official Review · Reviewer_wQek · 2025-06-30

**Clarity:** 3
**Significance:** 3
**Originality:** 3
**Rating:** 5
**Confidence:** 3

**Summary:**

This paper presents the first dynamic algorithm for non-monotone submodular maximization under general p-matchoid constraints. The authors develop a recursive filtering and sampling framework that maintains an approximate solution as elements are inserted or deleted. Their algorithm achieves a provable approximation guarantee and low amortized query complexity, improving prior results for special cases like cardinality constraints.

**Questions:**

1. The algorithm relies on random sampling to simulate monotone behavior, how sensitive is the performance to the sampling probability q?
2. Would a different choice affect approximation or update cost significantly?

**Ethical Concerns:**

["NO or VERY MINOR ethics concerns only"]

**Final Justification:**

My concerns have been fully addressed via author rebuttal.

**Limitations:**

Yes

**Quality:**

3

**Strengths And Weaknesses:**

Strengths:

1.The paper introduces the first dynamic algorithm for non-monotone submodular maximization under general p-matchoid constraints, achieving strong theoretical guarantees and improving upon prior results.

2.The recursive filtering and sampling framework is elegant and efficiently handles both insertions and deletions with provable amortized query complexity.

Weaknesses:

1.The theoretical query complexity includes high-degree polynomial terms, which may limit scalability in real-world applications.

2.The dynamic model is based on the oblivious adversarial assumption, and its robustness against adaptive attack scenarios has not been fully discussed.

---

> ### Author Rebuttal · Authors · 2025-07-31
>
> Thank you for the constructive and encouraging review. We greatly appreciate your positive assessment of the strength and elegance of our algorithm. Below, we address questions and concerns you raised.
>
> > The theoretical query complexity includes high-degree polynomial terms ...
>
> We note that the query complexity, although polynomial in $k$, is independent of $n$, the size of the ground set. This makes it practical in many real-world scenarios.
>
> > The dynamic model is based on the oblivious adversarial assumption, and its robustness against adaptive attack scenarios has not been fully discussed.
>
> The approximation guarantee of the algorithm does not rely on the oblivious adversarial assumption. However, our current analysis of the query complexity bounds is tailored to the oblivious setting
> and may not directly extend to scenarios against an adversary with full knowledge of the random bits used in the algorithm.
> However, we should note that this is a valid assumption in many real-world scenarios, and it is also consistent with prior work in fully dynamic submodular optimization, where, to the best of our knowledge, all existing results assume the same kind of adversarial setting.
> Nonetheless, robustness against adaptive adversaries is an interesting and important direction for future research in the field.
>
> > The algorithm relies on random sampling to simulate monotone behavior. How sensitive is the performance to the sampling probability q?
>
> The analysis of the algorithm is based on a fixed value for the parameter  $q = (p+\sqrt{p^2+p}+1)^{-1}$.
> We use equalities $\frac{1}{q} - 1 = (1 + c)p$ and $1 - q = \frac{1}{c}$ to get an approximation gurantee of $\frac{c}{(1+c)^2p}\left(\frac{1}{c} - \epsilon \right)$. These equalities hold for the said choice of $q$ and $c=\sqrt{1+\frac{1}{p}}$.
>
> We will clarify this earlier in the paper to avoid ambiguity.
>
> If you feel that our response has adequately addressed your concerns, we would appreciate it if you would possibly raise your score accordingly.

---

> > ### Author Response · Authors · 2025-08-03
> >
> > Dear Reviewer,
> >
> > We would like to kindly check whether you have any remaining questions or concerns. We would be happy to provide further clarification. If not, we would be grateful if you could consider updating your score in light of the rebuttal.

---

> > ### Comment · Reviewer_wQek · 2025-08-05
> > **Thanks for your response**
> >
> > The authors fully addressed my concerns. I'm glad to raise my rating.

---

### Official Review · Reviewer_fHaN · 2025-07-03

**Clarity:** 3
**Significance:** 3
**Originality:** 3
**Rating:** 4
**Confidence:** 3

**Summary:**

This paper investigates non-monotone submodular maximization in a fully dynamic setting with $p$-matchoid constraints. The main contributions are:
- A $(2p+2\sqrt{p(p+1)}+1+\epsilon)$-approximation algorithm with an expected amortized query complexity of $O(\epsilon^{-3}pk^4\log^2(k))$. This is the first theoretical advancement for this problem under $p$-matchoid constraints.
- As a byproduct, a $(5.82+\epsilon)$-approximation algorithm is proposed under size constraints with $O(\epsilon^{-3}k^4\log^2(k))$ oracle queries per update. This algorithm improves the best-known $(8+\epsilon)$ approximation ratio for this problem.

**Questions:**

- Empirical validation would significantly strengthen the paper's contributions. Especially the comparison of the $(5.82+\epsilon)$-approximation algorithm with the existing $(8+\epsilon)$-approximation algorithm under size constraints. Experiments under $p$-matchoid can also serve as reference points for future works.
- It might to better to shorten the first three sections and move some of the analysis into the analysis section.

**Ethical Concerns:**

["NO or VERY MINOR ethics concerns only"]

**Final Justification:**

I have read the rebuttal and other reviews. I will keep my score.

**Limitations:**

Yes

**Quality:**

3

**Strengths And Weaknesses:**

Strengths
- The study addresses an important and practically relevant problem in submodular optimization, with clearly motivated applications discussed such as feature selection.
- The paper provides the first approximation algorithm for non-monotone submodular maximization under $p$-matchoid constraints in dynamic settings. As a by-product, it also improves the approximation ratio for the special case of cardinality constraints, advancing the state of the art.
- The paper is clearly written and logically structured, making the technical contributions accessible to readers.

Weaknesses

Given the practical nature of dynamic settings, experimental validation on real-world or synthetic datasets would strengthen the paper’s impact.

---

> ### Author Rebuttal · Authors · 2025-07-31
>
> We thank the reviewer for the positive evaluation and for highlighting the importance and quality of our contributions.
> We are very pleased that you found our paper well-written and technically solid.
>
> We appreciate your helpful feedback and will incorporate your suggestions into our paper.
>
> Specifically, regarding empirical evaluations, while the primary focus of this work has been on the theoretical development, we fully agree that empirical evaluations would further enhance the impact of our work.
> For the special case of cardinality constraints, we have conducted experiments on real-world datasets (videos from YouTube and Open Video Project for the task of video summarization).
> We compared our algorithm’s query complexity and submodular value against the state-of-the-art $(8+\epsilon)$-approximation algorithm, and the results showcased the competitive performance of our algorithm. We will include these in the revised version of our paper.

---

### Official Review · Reviewer_Tfaj · 2025-07-03

**Clarity:** 3
**Significance:** 3
**Originality:** 2
**Rating:** 5
**Confidence:** 4

**Summary:**

This paper studies non-monotone submodular maximization with p-matchoid constraint, in the fully dynamic setting. In such setting, elements are inserted and deleted by an adversary, and the goal is to maintain an high-quality solution using few queries. Typically, the amortized number of queries should not depend polynomially in $n$, the length of the stream.

The authors propose an $\approx 4p$ approximation for $p$-matchoids, building on the fully-dynamic framework of Banihashem et al.

**Questions:**

None

**Ethical Concerns:**

["NO or VERY MINOR ethics concerns only"]

**Final Justification:**

I confirm my score

**Limitations:**

nothing to add here

**Paper Formatting Concerns:**

nothing to add here

**Quality:**

3

**Strengths And Weaknesses:**

- Submodular maximization is an important optimization task that has been studied intensively by the NeurIPS and ICML communities
- The fully dynamic setting is well motivated in practice
- Non-monotone submodular maximization naturally arises in applications
- p-matchoids are arguably the most general constraints in the submodular maximization literature
- the paper is fairly well written
- the authors improve on the state of the art for cardinality constraints (from 8 to 5.82, but at the cost of worse running time)
- The technical contribution is non-trivial, but not totally unexpected and original. The author's construction is heavily inspired by the versatile framework of Banihashem et al. and adapted to $p$-matchoids.

---

> ### Author Rebuttal · Authors · 2025-07-31
>
> We thank the reviewer for recognizing the importance of our problem and contributions. We are pleased that you found the paper well-written and of high quality, and we greatly appreciate the positive and affirming review.

---

### Official Review · Reviewer_uQeU · 2025-07-03

**Clarity:** 4
**Significance:** 3
**Originality:** 3
**Rating:** 5
**Confidence:** 3

**Summary:**

The paper studies dynamic maximization of submodular objective over a matchoid where single elements are inserted or removed dynamically. A 2-part algorithm is given; it builds an initial solution, and updates it as the ground set is modified. The paper includes its theoretical analysis, with findings pertaining to the query complexity and online approximation guarantees.

**Questions:**

Since $f$ can be non-monotone, the assertion that $\min_{X\in\mathcal{I}}f(X)=0$ would preserve the generality of the problem, while $f(\emptyset)=0$ would not. Do the results in this work require that the latter hold, or would a more careful application of notations suffice?

Algorithm 1: is $A_0$ assigned 0 or $\emptyset$?

**Ethical Concerns:**

["NO or VERY MINOR ethics concerns only"]

**Final Justification:**

The authors adequately addressed my concerns, which are minor. Given that the additional experimental results are unavailable at this time, I will keep my score.

**Limitations:**

yes

**Quality:**

4

**Strengths And Weaknesses:**

The paper is written clearly. The concepts and settings are explained in a way that accommodates unfamiliar readers. The method is explained well, and adequately contextualized by existing work.

The contribution is significant due to the breadth of the considered setting. General submodular functions are relevant to machine learning, and matchoids capture a wide range of independence systems. The proposed method also improves over an existing method under a simpler setting in approximation guarantees, at the cost of increased query complexity.

Due to the generality of the setting, sharpened results in popular special cases would be appreciated. Only one case is listed. There is a large body of work on monotone objectives. I believe the approximation bound in this work can be strengthened under monotonicity of $f$.

The algorithm contains some stochastic components, the most impactful of which is the RateSampling subroutine. It'd make the paper more self-contained to have its relevant probabilistic properties stated in Section 4.1.

The study lacks empirical investigation. It'd be nice to see the online approximation quality of the algorithm under various insertion/deletion patterns, or at least a direct comparison to the performances given by Banihashem et. al, 2023.

---

> ### Author Rebuttal · Authors · 2025-07-31
>
> Thank you so much for your positive feedback. We are glad you were satisfied with the clarity, significance, and quality of our paper. We address specific questions and comments below:
>
> > Since $f$ can be non-monotone, the assertion ... . Do the results in this work require that the latter hold, or would a more careful application of notations suffice?
>
>
> Thank you for the question. The stronger assumption is not required, and non-negativity of the function $f$ is sufficient for the analysis to hold, which, as you noted, preserves the generality of the problem. We have made proper adjustments to the notations accordingly.
>
> > Algorithm 1: is $A_0$ assigned 0 or $\emptyset$?
>
> The variable $A_0$ is initialized to $\emptyset$. Thanks for catching this.
>
> > It'd be nice to see the online approximation quality of the algorithm under various insertion/deletion patterns, or at least a direct comparison to the performances given by Banihashem et. al, 2023.
>
> We acknowledge that empirical results would provide further valuable insight. We have conducted experiments evaluating the performance of
> our algorithm for video summarization using datasets obtained from YouTube and the Open Video Project.
> For the special case of cardinality constraint, we benchmarked the total number of query calls and the submodular value of our solution, demonstrating competitive performance compared to [Banihashem et al., 2023].
> As per the guidelines, we are unable to provide these during the rebuttal period; however, we will incorporate them in the final version of the paper.

---

> > ### Comment · Reviewer_uQeU · 2025-08-05
> >
> > Thanks for addressing the questions. The setting for additional experimentation seems adequate, although I can't judge the results' merits. I will keep my score.

---

### Decision · Program_Chairs · 2025-09-17

**Decision:**

Accept (poster)

**Comment:**

This paper studies submodular maximization in the dynamic setting. The main result is an algorithm that maintains an approximate solution with a fast update time for non-monotone submodular maximization over a p-matchoid constraint. Previous dynamic algorithms were either for non-monotone functions under a cardinality constraint or for monotone functions under a matroid constraint. p-matchoid constraints are a general family of constraints that generalize matroids.
This paper achieves a strong result for a very general setting. The algorithm is technically involved and interesting (but does mostly build on ideas from previous work).